# BRIDGEDRIVE: DIFFUSION BRIDGE POLICY FOR CLOSED-LOOP TRAJECTORY PLANNING IN AUTONOMOUS DRIVING

**Shu Liu**[*,†]**, Wenlin Chen**[*,†]**, Weihao Li**[*]**, Zheng Wang**[*,†]**,**
**Lijin Yang, Jianing Huang, Yipin Zhang, Zhongzhan Huang, Ze Cheng, Hao Yang**[†]
Bosch (China) Investment Ltd. & Bosch Center for AI (BCAI)
[†]`{shu.liu2, wenlin.chen, david.wang3, kevin.yang}@cn.bosch.com`

## ABSTRACT

Diffusion-based planners have shown strong potential for autonomous driving by capturing multi-modal driving behaviors. A key challenge is how to effectively guide these models for safe and reactive planning in closed-loop settings, where the ego vehicle's actions influence future states. Recent work leverages typical expert driving behaviors (*i.e.*, anchors) to guide diffusion planners but relies on a truncated diffusion schedule that introduces an asymmetry between the forward and denoising processes, diverging from the core principles of diffusion models. To address this, we introduce *BridgeDrive*, a novel anchor-guided diffusion bridge policy for closed-loop trajectory planning. Our approach formulates planning as a diffusion bridge that directly transforms coarse anchor trajectories into refined, context-aware plans, ensuring theoretical consistency between the forward and reverse processes. BridgeDrive is compatible with efficient ODE solvers, enabling real-time deployment. We achieve state-of-the-art performance on the Bench2Drive closed-loop evaluation benchmark, improving the success rate by 7.72% and 2.45% over prior arts with PDM-Lite and LEAD datasets, respectively. Project page: https://github.com/shuliu-ethz/BridgeDrive.

## 1 INTRODUCTION

Closed-loop planning with reactive agents is a critical challenge in autonomous driving, which requires effective interaction with complex and dynamic traffic environments (Jia et al., 2024). Diffusion models have become a powerful paradigm for this task due to their ability to model complex, multi-modal distributions and incorporate flexible guidance (Liao et al., 2025; Zheng et al., 2025b; Yang et al., 2024; Xing et al., 2025). A key challenge, however, is to determine which sources of guidance information are most salient and how to integrate them effectively into these models to produce plans that are not only plausible but also safe and reactive in real-world driving conditions.

A promising source for guidance is to leverage typical human expert driving behaviors, often represented as coarse *anchor* trajectories, as they provide a strong prior for safe and sensible maneuvers, constraining the vast solution space. Recently, DiffusionDrive (Liao et al., 2025) implements this strategy by training a denoiser on a truncated diffusion schedule, starting from a noisy version of the anchor rather than pure Gaussian noise. While achieving state-of-the-art empirical performance, this approach introduces a theoretical inconsistency: its denoising process does not match the forward diffusion process that it is trained on, which diverges from the core principle of diffusion models and can lead to unpredictable behaviors and compromised performance.

To address this, we introduce *BridgeDrive*, a principled diffusion framework that integrates anchor-based guidance for autonomous driving planning using a theoretically sound diffusion bridge formulation. Instead of heuristically truncating the diffusion process, we formally define the planning task as learning a diffusion process that *bridges* the gap from a given coarse anchor trajectory to a refined, context-aware final trajectory plan. This formulation ensures that the forward and denoising processes are perfectly symmetric, allowing our model to learn a direct and robust transformation

---

* denotes equal contribution. Corresponding author: Shu Liu.

from anchors to final trajectories. By adhering to the principles of diffusion, our method fully leverages the expressive power of anchors for guidance while maintaining diffusion models' ability to represent diverse human-like driving behaviors. Furthermore, our approach is compatible with efficient ODE-based samplers, enabling real-time performance crucial for on-road deployment. Empirically, we achieve 74.99% and 89.25% success rate on the Bench2Drive closed-loop benchmark with PDM-Lite and LEAD datasets, respectively, outperforming previous SOTA by 7.72% and 2.45%.

## 2 PRELIMINARIES

### 2.1 AUTONOMOUS DRIVING PLANNING AND EVALUATION

The planning task in autonomous driving can be formulated as predicting future trajectories of the ego-vehicle based on raw sensor inputs. Conventionally, there are two trajectory representations (Renz et al., 2025): (1) **Temporal speed waypoints** $x := x^{\text{temp}} \in \mathbb{R}^{N_{\text{point}} \times 2}$, represent equal temporal-spaced (e.g., every 0.25 seconds) future coordinates of ego-vehicle, which inherently contain speed control information. (2) **Geometric path waypoints** $x := (x^{\text{geo}}, v) \in \mathbb{R}^{N_{\text{point}} \times 2} \times \mathbb{R}$, represent equal geometric-spaced (e.g., every 1 meter) future coordinate of ego-vehicle; for geometric path waypoints-based planning, the model needs to predict the speed $v$ of ego-vehicle. In this paper, we choose to use geometric path waypoints as our model output, which differs from DiffusionDrive Liao et al. (2025) where temporal speed waypoints are used. This design choice is based on prior works (Chitta et al., 2023; Zimmerlin et al., 2024) and our ablation study in Section 4.

Evaluation of autonomous driving can be broadly categorized into open-loop and closed-loop settings. The closed-loop setting is more challenging and can better reflect a policy's real-world planning capability, since the ego vehicle's decisions affect its own future states and those of the surrounding agents, creating a feedback loop that can amplify small prediction errors over time. To minimize the sim-to-real gap, closed-loop evaluation requires high-fidelity simulators to capture the interactions between the ego vehicle and its surrounding environment, which are typically both computationally expensive and time-consuming. CARLA (Dosovitskiy et al., 2017) has emerged as the most widely used platform, with a series of benchmarks building on top of it, such as CARLA Leaderboard, Longest6 (Chitta et al., 2023), and Bench2Drive (Jia et al., 2024). Interestingly, existing methods that achieve near-perfect results on open-loop datasets, such as NavSim (Dauner et al., 2024) or nuScenes (Caesar et al., 2019), still struggle to achieve comparable performance under closed-loop evaluation (Li et al., 2024b; Liao et al., 2025; Fu et al., 2025; Renz et al., 2025). This discrepancy emphasizes the inherent difficulty of closed-loop planning and highlights the need for more robust methods to handle the complexities of dynamic, interactive traffic environments.

### 2.2 DIFFUSION MODELS

Diffusion models (Sohl-Dickstein et al., 2015; Ho et al., 2020; Song et al., 2021a;b; Karras et al., 2022) generate data $x_0 \sim p_d(x_0)$ from pure Gaussian noise $x_T \sim p(x_T) := \mathcal{N}(x_T|0, \sigma_{\max}^2 I)$ by reverting a forward diffusion process. Mathematically, the forward diffusion process, which gradually corrupts data into noise, can be defined by a linear SDE (Song et al., 2021b):

$$\mathrm{d}x_t = f(t)x_t \mathrm{d}t + g(t)\mathrm{d}w_t, \quad x_0 \sim p_d, \tag{1}$$

where $t \in [0, T]$ denotes the diffusion timestep, $f : [0, T] \to \mathbb{R}$ is the linear drift coefficient, $g : [0, T] \to \mathbb{R}_+$ is the diffusion coefficient function, and $w_t \in \mathbb{R}^d$ is a standard Brownian motion. It turns out that this linear SDE owns a Gaussian transition kernel $q(x_t|x_0) = \mathcal{N}(x_t|\alpha_t x_0, \sigma_t^2 I)$, where $\alpha_t = \exp\left(\int_0^t f(s)\mathrm{d}s\right)$ and $\sigma_t^2 = \alpha_t^2 \int_0^t \frac{g(s)^2}{\alpha_s^2}\mathrm{d}s$ are the noise schedules (Kingma et al., 2021). The forward SDE defines a series of marginals densities $\{q(x_t)\}_{t \in [0,T]}$ along the diffusion path, where $q(x_t) = \int q(x_t|x_0)p_d(x_0)\mathrm{d}x_0$. Since $q(x_T) \approx p(x_T)$ for sufficiently large $T$, we can generate data $x_0 \sim p_d(x_0)$ by transforming a noise sample $x_T \sim p(x_T)$ through a probability flow ODE (PF-ODE) (Song et al., 2021b):

$$\frac{\mathrm{d}x_t}{\mathrm{d}t} = f(t)x_t - \frac{g(t)^2}{2}\nabla_{x_t} \log q(x_t), \tag{2}$$

which shares identical marginal densities $\{q(x_t)\}_{t \in [0,T]}$ as the forward SDE. The score function $\nabla_{x_t} \log q(x_t)$ in Eq. (2) can be approximated by $\nabla_{x_t} \log q(x_t) \approx (\alpha_t x_\theta(x_t, t) - x_t)/\sigma_t^2$ (Vincent,

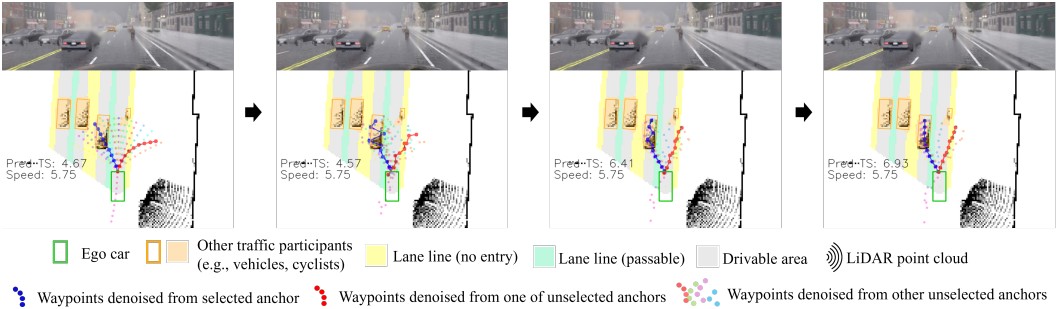

Figure 1: BridgeDrive denoising process ($t = T \to 0$ from left to right). A classifier selects the best anchor (blue dots) based on traffic context. The denoising proceeds from anchor $x_T$ (leftmost) to final trajectory $x_0$ (rightmost). Red dots show the denoising of an unselected (negative) anchor, highlighting the importance of correct anchor guidance in diffusion. The classifier achieves high accuracy (nearly 100%) in practice; its impact on planning is analyzed in the Appendix D.1.

2011), where the denoiser $x_\theta(x_t, t)$ is parameterized by a neural network and learned by minimizing the mean squared denoising error (Karras et al., 2022):

$$\min_\theta \ \mathbb{E}_{p(t)p_d(x_0)q(x_t|x_0)} \left[ w(t) \| x_\theta(x_t, t) - x_0 \|^2 \right].\tag{3}$$

For conditional generation, the denoiser $x_\theta(x_t, t, z)$ takes in an extra conditional variable $z$, which corresponds to the conditional score function $\nabla_{x_t} \log q(x_t|z) \approx (\alpha_t x_\theta(x_t, t, z) - x_t)/\sigma_t^2$. Furthermore, Ho & Salimans (2021) propose to linearly interpolate between $\nabla_{x_t} \log q(x_t|z)$ and $\nabla_{x_t} \log q(x_t)$ with a hyperparameter to adjust the guidance strength of the conditional information.

### 2.3 DiffusionDrive with Truncated Diffusion

DiffusionDrive (Liao et al., 2025) is a diffusion planner based on temporal speed waypoints, which leverages a truncated diffusion schedule and a fixed set of $K$-means clustered anchor trajectories $\mathcal{Y} = \{y^i\}_{i=1}^{N_{\text{anchor}}}$ that represent typical human driving behaviors. The truncated forward diffusion process adds a small amount of noise to each anchor until $t = T_{\text{trunc}} \ll T$ to obtain a set of noisy anchors $\{y_{T_{\text{trunc}}}^i\}_{i=1}^{N_{\text{anchor}}}$. The truncated denoising process starts from noisy anchors at $t = T_{\text{trunc}}$. Given conditional information $z$ (e.g., sensor inputs and target point), a neural network $x_\theta(\{y_t^i\}_{i=1}^{N_{\text{anchor}}}, t, z)$ is trained to predict the best anchor and output a denoised trajectory from the noisy version of the best anchor. The denoised trajectory is then used to compute the score function for denoising.

However, as discussed in the previous section, the learned denoising process of diffusion models must revert the forward diffusion process. Although DiffusionDrive demonstrates strong empirical performance, it utilizes a truncated diffusion schedule where the forward diffusion process adds noise to anchor trajectories and the denoising process attempts to recover the ground-truth trajectories. This design choice creates an asymmetry between its forward and denoising processes, framing the model's task as regressing from noisy anchors to ground-truth trajectories, rather than as a reversal of the forward diffusion process. A conceptual comparison between a standard full diffusion model, DiffusionDrive, and BridgeDrive is illustrated in Appendix B.5 and Fig. 8.

## 3 BridgeDrive: Diffusion Bridge Policy for Trajectory Planning

To ensure the symmetry between the forward and backward processes of anchor-based diffusion planners, we propose a novel diffusion bridge policy, *BridgeDrive*, which provides a principled diffusion framework that leverages the powerful inductive biases of anchor-based guidance, while ensuring that the symmetry between the forward and denoising processes is maintained.

### 3.1 Anchor Construction for Geometric Path Waypoints

Anchors $\mathcal{Y} = \{y^i\}_{i=1}^{N_{\text{anchor}}}$ are pre-defined, high-priority trajectories that represent typical human expert driving behaviors. They form a discrete set of atomic building blocks that planners use to

construct solutions, which can dramatically reduce planning complexity, enforce safety constraints, improve robustness to dynamic environments, and align planning with task objectives (Chai et al., 2020; Chen et al., 2024; Li et al., 2024b). Our anchor definition is slightly different from Liao et al. (2025) since our model outputs geometric path waypoints as discussed in Section 2.1. Specifically, each anchor is formulated as[1] $y := (x_y^{\text{geo}}, v_y) \in \mathbb{R}^{N_{\text{point}} \times 2} \times \mathbb{R}$, where $x_y^{\text{geo}} \in \mathbb{R}^{N_{\text{point}} \times 2}$ represents a series of coordinates of future path, $N_{\text{point}}$ is the geometric prediction horizon, and $v_y$ denotes the anchor speed. Each anchor $y$ is defined as a $K$-means clustering center of the training set (both the trajectory $x_y^{\text{geo}}$ and speed $v_y$ participate in the clustering process together). All values are normalized to the ego-vehicle coordinate system.

---

**Algorithm 1** BridgeDrive Training (ours)

1: Initialize $\theta$        # denoiser parameter
2: **repeat**
3:     $x, y, z \sim p_d(x, y, z)$
4:        # $x$: GT traj, $y$: anchor, $z$: guidance
5:     $t \sim p(t)$        # $t \in [0, T]$
6:     $\epsilon \sim \mathcal{N}(0, I)$      # random noise
7:     $x_t = a_t y + b_t x + c_t \epsilon$   # noisy trajectory
8:     Update $\theta$ with the gradient

$$w(t)\nabla_\theta \|x_\theta(x_t, t, y, z) - x\|^2$$

9: **until** convergence
10: **return** $\theta$

**Algorithm 2** DiffusionDrive Training

1: Initialize $\theta$        # denoiser parameter
2: **repeat**
3:     $x, y, z \sim p_d(x, y, z)$
4:        # $x$: GT traj, $y$: anchor, $z$: guidance
5:     $t \sim p_{\text{trunc}}(t)$      # $t \in [0, T_{\text{trunc}}]$
6:     $\epsilon \sim \mathcal{N}(0, I)$      # random noise
7:     $y_t = \alpha_t y + \sigma_t \epsilon$      # noisy anchor
8:     Update $\theta$ with the gradient

$$w(t)\nabla_\theta \|x_\theta(y_t, t, z) - x\|^2$$

9: **until** convergence
10: **return** $\theta$

---

## 3.2   A Generative Paradigm for Anchor-Guided Diffusion Policy

To incorporate anchors into diffusion models in a principled way, we propose to factorize the joint distribution of the ground-truth trajectory $x$, anchor $y$, and guidance information $z$ as

$$p_d(x, y, z) = p_d(x|y, z)p_d(y|z)p_d(z). \tag{4}$$

This factorization defines a two-step generative process. First, for a driving scene $z \sim p_d(z)$, we sample an anchor $y \sim p_d(y|z)$ given the scene information in $z$ (e.g., BEV, agent/map queries, and a target point). Then, the planned trajectory $x \sim p_d(x|y, z)$ is generated according to the guidance of the chosen anchor $y$ and scene information $z$.

We propose to parameterize the conditional planning distribution $p_d(x|y, z)$ with a conditional diffusion bridge model $p_\theta(x_t|x_T, z)$, which constructs a diffusion bridge (Zhou et al., 2024; Zheng et al., 2025a) between the ground-truth trajectory $x_0 := x$ and anchor $x_T := y$ (Doob et al., 1984):

$$\mathrm{d}x_t = f(t)x_t\mathrm{d}t + g(t)^2 \nabla_{x_t} \log q(x_T|x_t) + g(t)\mathrm{d}w_t, \quad x_0 \sim p_d, \ x_T = y, \tag{5}$$

where $t \in [0, T]$ denotes the diffusion timestep, the definitions of $f(t), g(t)$ follow those in Eq. (1), and $\nabla_{x_t} \log q(x_T|x_t) = \nabla_{x_t} \log q(x_t|x_0, x_T) - \nabla_{x_t} \log q(x_t|x_0)$. It turns out that Eq. (5) also owns an analytical Gaussian transition kernel for any given trajectory $x_0$ and anchor $x_T$:

$$q(x_t|x_0, x_T) = \mathcal{N}(x_t|a_t x_T + b_t x_0, c_t^2 I), \tag{6}$$

$$a_t = \alpha_t \gamma_t^2/\alpha_T, \quad b_t = \alpha_t(1 - \gamma_t^2), \quad c_t^2 = \sigma_t^2(1 - \gamma_t^2), \tag{7}$$

where $\alpha_t = \exp\left(\int_0^t f(s)\mathrm{d}s\right)$, $\sigma_t^2 = \alpha_t^2 \int_0^t \frac{g(s)^2}{\alpha_s^2}\mathrm{d}s$, and $\gamma_t = \frac{\alpha_T \sigma_t}{\alpha_t \sigma_T}$ (Zheng et al., 2025a), which defines a diffusion bridge $x_t = a_t x_T + b_t x_0 + c_t \epsilon_t$ that interpolates between $x_0$ and $x_T$ with added Gaussian noise $c_t \epsilon_t$. Zhou et al. (2024) show that there exists a PF-ODE that shares identical marginal densities $\{q(x_t|x_T)\}_{t \in [0,T]}$ as the forward diffusion bridge SDE in Eq. (5):

$$\frac{\mathrm{d}x_t}{\mathrm{d}t} = f(t)x_t - g(t)^2 \left( \frac{\nabla_{x_t} \log q(x_t|x_T, z)}{2} - \nabla_{x_t} \log q(x_T|x_t) \right), \tag{8}$$

---

[1] The subscript in $x_y, v_y$ indicates that the trajectory and its speed *belongs to an anchor*, which differs from an ordinary trajectory $x$ and its speed $v$. The superscript index $i$ in $y^i$ is omitted for notation simplicity.

---

**Algorithm 3** BridgeDrive Planning (ours)

---

1: $z \sim p_d(z)$        # sample a driving scene
2: $x_T = h_\phi(z, \mathcal{Y})$        # predict the optimal anchor
3: **for** $i = N, \cdots, 1$ **do**        # discretize timesteps into $T \equiv t_N < \cdots < t_1 < t_0 \equiv 0$
4:     $\hat{x}_{0|t_i} = x_\theta(x_{t_i}, t_i, x_T, z)$        # compute the denoised mean trajectory
5:     $\hat{s}_{t_i} = (a_{t_i} x_T + b_{t_i} \hat{x}_{0|t_i} - x_{t_i})/c_{t_i}^2$        # compute the score function
6:     $d_{t_i} = f(t_i)x_{t_i} - g(t_i)^2 \left( \hat{s}_{t_i}/2 - \nabla_{x_{t_i}} \log q(x_T|x_{t_i}) \right)$        # compute the derivative $dx_t/dt$
7:     $x_{t_{i-1}} = \text{ODESolverStep}(x_{t_i}, d_{t_i}, t_i, t_{i-1})$        # simulate the BridgeDrive PF-ODE
8: **end for**
9: **return** $x_0$

---

which allows us to translate an anchor $x_T$ to a planned trajectory $x_0$ given the driving scene $z$. To simulate this PF-ODE, we need to approximate the score function $\nabla_{x_t} \log q(x_t|x_T, z)$ for the conditional diffusion bridge model. In the next section, we will introduce our training and planning algorithms for this diffusion bridge policy.

## 3.3 TRAINING AND PLANNING ALGORITHMS

In our diffusion bridge planner, each diffusion bridge is constructed between a ground-truth trajectory $x_0 := x$ and the nearest anchor $x_T := y \in \mathcal{Y}$ to it. During training, we fit a neural network denoiser $x_\theta(x_t, t, x_T, z)$ to predict the denoising mean $\hat{x}_{0|t} \approx \mathbb{E}[x_0|x_t, x_T, z]$ given noisy trajectory $x_t \sim q(x_t|x_0, x_T)$ at timestep $t$, the nearest anchor $x_T = y$ to $x_0$, and the conditional information $z$ for the driving scene. This denoiser is trained by minimizing the mean squared denoising error:

$$\min_\theta \ \mathbb{E}_{p(t)p_d(x_0, x_T, z)q(x_t|x_0, x_T)} \left[ w(t) \| x_\theta(x_t, t, x_T, z) - x_0 \|^2 \right]. \tag{9}$$

Our training algorithm is summarized in Algorithm 1. Notice that our forward and reverse diffusion paths both result in the end point $x_0 = x$ since $a_0 = c_0 = 0$ and $b_0 = 1$, ensuring that the denoiser is trained to reverse the forward diffusion process. On the other hand, in DiffusionDrive (Liao et al., 2025) (Algorithm 2), for all $\alpha_t$ and $\sigma_t$, the noisy anchor $y_t$ deviates from $x$, failing to adhere to the symmetry requirement of diffusion models. Also, the training procedure of BridgeDrive is simulation-free, which allows us to efficiently train the denoiser without simulating the forward SDE in Eq. (5) or the PF-ODE in Eq. (8). In addition, since the ground-truth trajectory $x_0$ is not available for computing the nearest anchor $y$ at inference time, we also train a classifier $h_\phi(z, \mathcal{Y})$ to predict the nearest anchor $y$ to $x_0$ given $z$ with the cross entropy loss.

Similar to standard diffusion models, the trained denoiser $x_\theta(x_t, t, x_T, z)$ can be used to approximate the conditional score function for our conditional diffusion bridge model (Zheng et al., 2025a):

$$\nabla_{x_t} \log q(x_t|x_T, z) \approx \frac{a_t x_T + b_t x_\theta(x_t, t, x_T, z) - x_t}{c_t^2}. \tag{10}$$

Our planning algorithm is summarized in Algorithm 3 and depicted in Fig. 2. Specifically, for a given driving scene $z$, we first use the classifier $h_\phi(z, \mathcal{Y})$ to choose an anchor $y \in \mathcal{Y}$, which is the starting point $x_T = y$ of the denoising process in our diffusion bridge planner. Then, we iteratively compute the denoised mean trajectory $\hat{x}_{0|t}$ using our denoiser $x_\theta(x_t, t, x_T, z)$, calculate the score function $\hat{s}_t$ using Eq. (10), and simulate the PF-ODE in Eq. (8) with the score $\hat{s}_t$ using a numerical ODE solver. Although image diffusion models use higher-order ODE solvers (Karras et al., 2022; Lu et al., 2022) to accelerate sampling, we find that first-order methods, such as the DDIM sampler (Song et al., 2021a), are sufficient for the planning task with minimal number of function evaluations. Fig. 1 visualizes the denoising process of BridgeDrive for an example driving scenario.

## 3.4 MODEL ARCHITECTURE

Our model consists of three major components: perception module, denoiser, and classifier. Implementation and training details are provided in Appendix C.

**Perception Module.** The perception module extract useful features from lidar, front camera image, and target point for the downstream diffusion planner. We use a pre-trained perception backbone

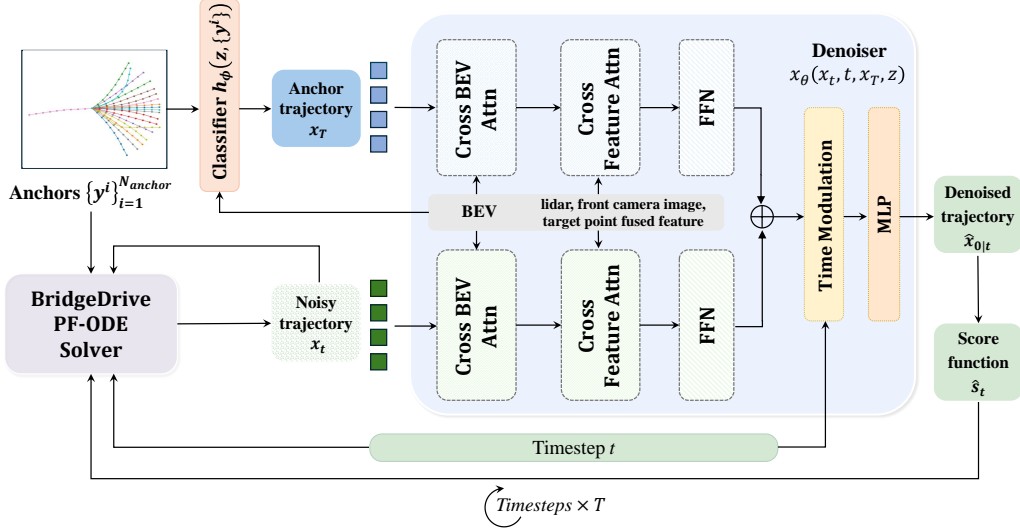

Figure 2: Diagram for the planning procedure of BridgeDrive in Algorithm 3. The model architecture of the neural network denoiser $x_\theta(x_t, t, x_T, z)$ is detailed in the light blue box.

from TransFuser++ (Jaeger et al., 2023) to obtain BEV segmentation, bounding boxes of traffic participants, general traffic information (e.g., stop signs and traffic lights), and fused features from the inputs. The output of the perception module is denoted as $z$ and will be used as the conditional guidance information in the denoiser module $x_\theta(x_t, t, x_T, z)$.

**Denoiser.** The architecture of the denoiser $x_\theta(x_t, t, x_T, z)$ is illustrated in the light blue box in Fig. 2. For a noisy trajectory $x_t$ at timestep $t$ and its corresponding anchor $x_T = y$, we first interact them with BEV via deformable spatial cross-attention modules. Subsequently, cross-attention with fused features from lidar, front camera, and target point is applied. The resulting feature vectors are further processed by feed-forward networks (FFNs), and their concatenation is modulated by the timestep $t$. Finally, an MLP network is employed to predict the denoised mean trajectory $\hat{x}_{0|t}$.

**Anchor Classifier.** The classifier $h_\phi(z, \mathcal{Y})$ employs cross BEV attention module and cross feature attention module between $z$ and all anchors $\mathcal{Y}$, followed by an FFN which outputs the probability that each anchor $y^i \in \mathcal{Y}$ should be used for trajectory generation. We select the anchor $y$ with the highest probability as the input $x_T$ to the denoiser $x_\theta(x_t, t, x_T = y, z)$. Note that the classifier only needs to be run once prior to the iterative denoising process.

## 4 EXPERIMENTS

### 4.1 CLOSED-LOOP EVALUATION ON BENCH2DRIVE

**Benchmark.** In this paper, we mainly focus on closed-loop evaluation since it simulates dynamic traffic conditions which can better reflect a policy's real-world planning capability. Bench2Drive (Jia et al., 2024) is a widely used closed-loop evaluation benchmark (Jaeger et al., 2023; Jia et al., 2025; Fu et al., 2025; Renz et al., 2025), which contains 220 routes for evaluation under the CARLA Leaderboard 2.0 protocol for end-to-end autonomous driving. Each route is around 150 meters in length and contains a specific driving scenario, which allows for a detailed assessment of autonomous driving systems' proficiency in different driving skills.

**Dataset.** While Bench2Drive provides an official training set, empirical studies (Zimmerlin et al., 2024; Renz et al., 2025; Fu et al., 2025) showed that official dataset collected by (Li et al., 2024a) leads to suboptimal performance. Therefore, data augmentation and cleansing scheme are applied to enhance the performance. For example, ORION (Fu et al., 2025) generated Visual Question Answering (VQA) to enhance their Vision-Language-Action (VLA) models' capability, such as scene description, behavior description, meta-driving decision and reasoning, and recall of essential histor-

Table 1: Comparison between BridgeDrive and previous baselines on Bench2Drive. Our method shows SOTA performance on both Driving Score (DS) and Success Rate (SR). Notably, by using a principled diffusion bridge model, our method achieves significant improvements over previous diffusion baselines (including those with prior knowledge from VLA), demonstrating the effectiveness of the diffusion module in the autonomous driving task when following our paradigm as discussed in Section 3.2. A potential avenue to further improve our method is to integrate prior knowledge from VLA, which is left as future work.

| Method | Expert | VLA | Diffusion | DS | SR(%) |
|---|---|---|---|---|---|
| TCP-traj* (Wu et al., 2022) | Think2Drive | ✗ | ✗ | 59.90 | 30.00 |
| UniAD-Base (Hu et al., 2023) | Think2Drive | ✗ | ✗ | 45.81 | 16.36 |
| VAD (Jiang et al., 2023) | Think2Drive | ✗ | ✗ | 42.35 | 15.00 |
| DriveTransformer (Jia et al., 2025) | Think2Drive | ✗ | ✗ | 63.46 | 35.01 |
| ORION (Fu et al., 2025) | Think2Drive | ✔ | ✗ | 77.74 | 54.62 |
| ORION diffusion (Fu et al., 2025) | Think2Drive | ✔ | ✔ | 71.97 | 46.54 |
| DiffusionDrive$^{temp}$ (Liao et al., 2025) | PDM-Lite | ✗ | ✔ | 77.68 | 52.72 |
| DiffusionDrive$^{geo}$ (Liao et al., 2025) | PDM-Lite | ✗ | ✔ | 80.79 | 58.18 |
| SimLingo (Renz et al., 2025) | PDM-Lite | ✔ | ✗ | 85.07 | 67.27 |
| TransFuser++ (Zimmerlin et al., 2024) | PDM-Lite | ✗ | ✗ | 84.21 | 67.27 |
| BridgeDrive (ours) | PDM-Lite | ✗ | ✔ | 87.99(+2.92) | 74.99(+7.72) |

ical information. Chitta et al. (2023) and Zimmerlin et al. (2024) use PDM-Lite (Beißwenger, 2024; Sima et al., 2025), an open source rule-based expert to collect ground-truth trajectories for imitation learning. SimLingo (Renz et al., 2025) generates additional driving data and applies intricate filtration on training routes of Chitta et al. (2023) and the official CARLA LB 2.0 routes. We use the datasets proposed by Zimmerlin et al. (2024). The dataset actively filters for critical change frames and refines expert behavior, thereby focusing on high-quality decision-making moments while reducing its size, which improves training efficiency and strengthens the model's learning in crucial driving scenarios. Preliminary results of BridgeDrive on the LEAD dataset (Nguyen et al., 2026) are reported in Section 4.3 and Appendix E, where BridgeDrive achieves a new SOTA with $89.25\%$ success rate and 96.34 driving score.

**Baselines.** We compare against the following baselines. *TCP-traj* (Wu et al., 2022) is a monocular camera-based method that jointly learns planning and direct control with a situation-based fusion. *UniAD* (Hu et al., 2023) is a unified end-to-end framework that integrates full-stack driving tasks through query-based interfaces. *VAD* (Jiang et al., 2023) is an end-to-end vectorized paradigm that models driving scenes with vectorized representations. *DriveTransformer* (Jia et al., 2025) employs task parallelism, sparse representation, and streaming to enable efficient cross-task knowledge transfer and temporal fusion. *ORION* (Fu et al., 2025) integrates a QT-Former for history aggregation, a reasoning large language model (LLM), and a VAE for planning. *Simlingo* (Renz et al., 2025) leverages VLA and achieves current SOTA performance on Bench2Drive. *TransFuser++* (Chitta et al., 2023) (Zimmerlin et al., 2024) (Jaeger et al., 2023) ranks second in the 2024 CARLA challenge and first on the Bench2Drive test routes. In addition, we adapt *DiffusionDrive* to Bench2Drive benchmark (denoted as DiffusionDrive$^{temp}$). Adaptation details are provided in Appendix C.2.

**Main results.** Evaluation results on Bench2Drive benchmark are demonstrated in Table 1. For all models that exceeded previous SOTA result, we performed experiments with three random seeds to ensure reproducibility. The performance of our work significantly exceeds that of all previous work. In particular, our BridgeDrive outperform the SimLingo (Renz et al., 2025), the latest SOTA, by +2.92 and +7.72% in driving score and success rate, respectively. Moreover, BridgeDrive exhibits outstanding multi-ability capability, especially in Merging (+11.17), and Traffic Sign (+7.02), resulting in overall improvement by +6.12 than SOTA, as shown in Table 6 in the Appendix. However, BridgeDrive demonstrates suboptimal performance in the Comfortness and Give Way metrics, suggesting a tendency toward frequent or poorly timed braking. This outcome may imply that our model

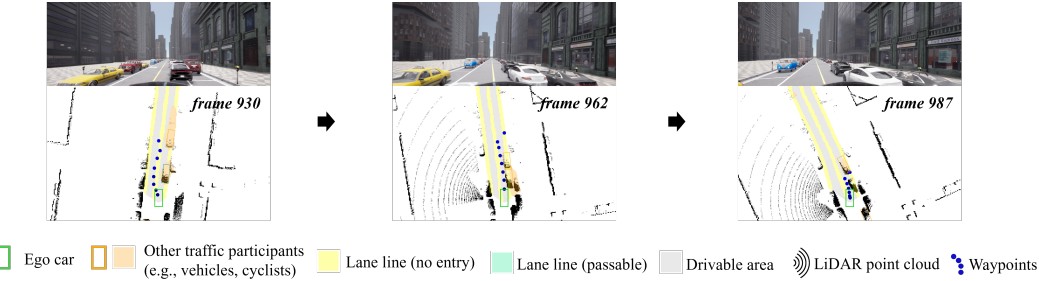

Figure 3: A consecutive three frames of a sample Bench2Drive scene, overtaking maneuver performed by **BridgeDrivetemp**. The ego car exhibited deficiencies in overtaking maneuver coordination and speed control, which directly led to a collision with the white vehicle. For video demonstration, please refer to supplementary materials.

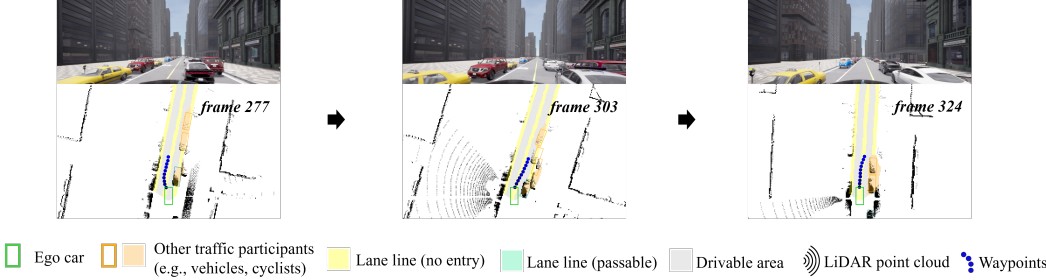

Figure 4: On the same scene as in Fig. 3, overtaking maneuver performed by **BridgeDrivegeo**. The ego vehicle adapts its planning to overtake a sequence of parked cars. For video demonstration, please refer to supplementary materials.

prioritizes safety considerations, potentially at the expense of passenger comfort. Furthermore, the inference speed of BridgeDrive is suitable for real-time deployment, as detailed in Table 7.

## 4.2 Ablation Study and Qualitative Analysis for the Closed-Loop Setting

The primer focus of this paper is on the design and study of diffusion models for trajectory planning; therefore, we prioritize the most vital aspects that could influence the performance, namely 1) what kind of trajectory representation is more compatible with diffusion model; 2) how our diffusion bridge policy with anchor guidance differs from other diffusion planners. Further ablation study results on the influence of anchors and classifiers are provided in Appendix D.

**Effectiveness of the representation of geometric path waypoints.** To account for the influence of different representations of the trajectory, namely the temporal speed waypoints vs. geometric path waypoints, we implement these two configurations for each version of diffusion models, denoted as *temp* and *geo*, respectively. It should be noted that, for DiffusionDrivegeo, all modules remain identical to those of our method except for the diffusion part to ensure a fair comparison. The results are compared in Table 2. It can be seen that the representation of geometric path waypoints outperforms their temporal counterpart, with an improvement of +5.46%, +4.09%, +15.09% in the success rate for DiffusionDrive, Full Diffusion, BridgeDrive, respectively. We argue the main reasons for this are as follows. 1) Temporal waypoints encode speed control information in the spacing between subsequent waypoints. Such an encoding is ambiguous and difficult to generalize. For example, for overtaking maneuvers with different speeds, geometric waypoints only require a model to learn the similar geometric pattern of driving path plus a varying speed scalar. In comparison, the generalization of temporal waypoints require a model to stretch spacing between waypoints to account for different speeds. 2) Geometric waypoints are more compliant with route topology and is therefore less likely to violate route lane constrain; similar arguments are also provided in Jaeger et al. (2023). A comparison between two kinds of waypoints is provided in Fig. 3 and Fig. 4.

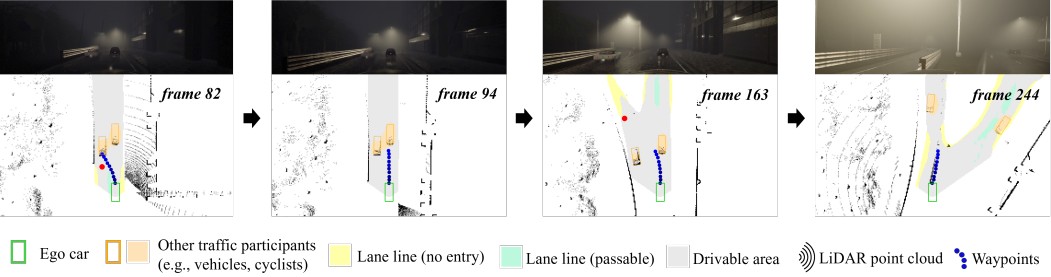

Figure 5: **Full Diffusion model** in a consecutive four frames of a sample Bench2Drive scene, **failing** to adhere to the target time window for lane-changing maneuvers, which consequently led to a collision with the road barrier. For video demonstration, please refer to supplementary materials.

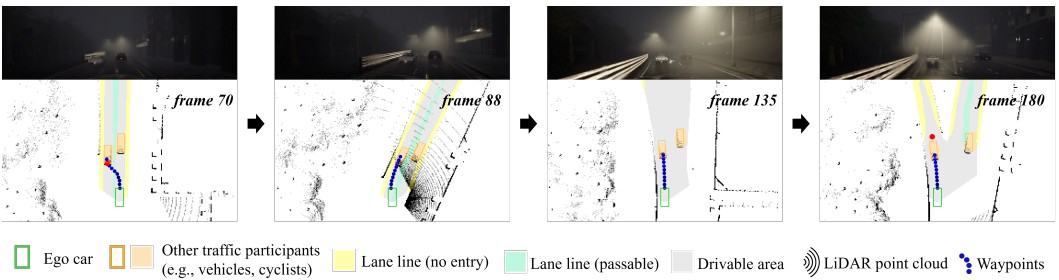

Figure 6: **BridgeDrive** on the same scene as in Fig. 5, **achieving** timely lane changing due to anchor guidance and successfully navigated through the road fork. For video demonstration, please refer to supplementary materials.

**Advantages of BridgeDrive.** As illustrated in Table 2, benefiting from the multi-modality of diffusion models, both Full Diffusion[geo] and BridgeDrive[geo] outperform DiffusionDrive[geo] by a large margin. In addition, compared with full diffusion, BridgeDrive further leverages anchor information to guide its diffusion process. This is of particular importance when facing ambiguous situations. An example is visualized in Fig. 5 and Fig. 6. In this case, the target point for lane change is given in a short distance ahead of the ego vehicle. Due to inherent inertial of the ego car, it is unlikely for full diffusion to change lane. Therefore, the ego car kept traveling in a straight path and missed the target point; subsequently, the ego car was unable to make a sharp turn to the left lane and hit the road barrier. In comparison, BridgeDrive, under the strong guidance of the anchor, was able to strictly follow the target point and entered the correct lane at the road fork.

Table 2: Ablation study for the effects of temporal and geometric path waypoints for Diffusion-Drive, full diffusion, and BridgeDrive. All methods use identical expert and modules except for the diffusion part. Our BridgeDrive[geo] achieves SOTA DS and SR, prioritizing safety over Comfortness.

| Configuration | Principled | Anchor | DS | SR(%) | Efficiency | Comfortness |
|---|---|---|---|---|---|---|
| DiffusionDrive[temp] | ✘ | ✔ | 77.68 | 52.72 | 248.18 | 24.56 |
| DiffusionDrive[geo] | ✘ | ✔ | 80.79 | 58.18 | 245.34 | 15.49 |
| Full Diffusion[temp] | ✔ | ✘ | 79.75 | 58.18 | 246.31 | 24.42 |
| Full Diffusion[geo] | ✔ | ✘ | 83.85 | 67.27 | 238.90 | 21.40 |
| BridgeDrive[temp] | ✔ | ✔ | 81.97 | 59.90 | 243.88 | 22.61 |
| BridgeDrive[geo] | ✔ | ✔ | 87.99 $\pm 0.67$ | 74.99 $\pm 1.35$ | 236.49 $\pm 2.32$ | 20.98 $\pm 0.74$ |

Table 3: Performance of BridgeDrive adapted to LEAD.

| Method | Expert | DS | SR(%) | Effi. | Comfort. |
|---|---|---|---|---|---|
| TFv6 (Nguyen et al., 2026) | LEAD | $95.2 \pm 0.3$ | $86.8 \pm 0.7$ | N/A | N/A |
| BridgeDrive (ours) | LEAD | $\mathbf{96.34} \pm 0.55$ | $\mathbf{89.25} \pm 0.50$ | $\mathbf{202.92} \pm 3.27$ | $\mathbf{23.24} \pm 1.06$ |

Table 4: Performance comparison on planning-oriented NAVSIM navtest split.

| Model | NC | DAC | TTC | Comf. | EP | PDMS |
|---|---|---|---|---|---|---|
| VADv2-$V_{8192}$ (Chen et al., 2024) | 97.2 | 89.1 | 91.6 | 100 | 76.0 | 80.9 |
| Hydra-MDP-$V_{8192}$-W-EP (Li et al., 2024b) | 98.3 | 96.0 | 94.6 | 100 | 78.7 | 86.5 |
| DiffusionDrive (reported in Liao et al. (2025)) | 98.2 | 96.2 | 94.7 | 100 | 82.2 | 88.1 |
| DiffusionDrive (reproduced with a different seed) | 98.2 | 95.9 | 94.3 | 100 | 81.9 | 87.6 |
| BridgeDrive (ours) | 98.2 | 96.1 | 94.5 | 100 | 82.3 | 88.0 |

## 4.3 GENERALIZATION TO LEAD DATASET

We further evaluate BridgeDrive on the LEAD dataset (Nguyen et al., 2026), which provides a refined expert policy designed to mitigate Learner–Expert Asymmetry in CARLA. Adaptation details are provided in Appendix E. As shown in Table 3, BridgeDrive trained on LEAD achieves a success rate of 89.25% and a driving score of 96.34, surpassing the LEAD baseline (TFv6) by +2.45% SR and +1.14 DS, with a mean multi-ability score of 82.22 (Table 12). These results demonstrate that BridgeDrive generalizes well across different training sets.

## 4.4 OPEN-LOOP EVALUATION ON NAVSIM

We additionally evaluate our method on NAVSIM using the metrics from DiffusionDrive (Liao et al., 2025), summarized by the PDM score (PDMS), a weighted composite of no at-fault collisions (NC), drivable area compliance (DAC), time-to-collision (TTC), comfort (Comf.), and ego progress (EP). As shown in Table 4, BridgeDrive achieves competitive performance on NAVSIM. We note that NAVSIM follows an open-loop evaluation protocol with non-reactive agents, where agent states are reset to the ground-truth at each step; as a result, this setting may underemphasize long-horizon error accumulation and interaction effects.

For this reason, we place primary emphasis on closed-loop evaluation (i.e., Bench2Drive), which more directly measures planning under feedback. This feedback is shared by a growing body of work advocating closed-loop assessment (Chitta et al., 2023; Zheng et al., 2025b; Caesar et al., 2021; Yang et al., 2025; Jia et al., 2024) and can amplify small errors across a rollout, as discussed in Section 2.1. BridgeDrive improves over prior methods under this more demanding evaluation protocol as shown in previous sections.

## 5 CONCLUSIONS AND FUTURE WORK

We presented BridgeDrive, an end-to-end autonomous driving solution based on diffusion bridge policy. Our method provides a principled diffusion framework incorporating anchor guidance, outperforming prior work by 7.72% and 2.45% in success rate on Bench2Drive with PDM-Lite and LEAD datasets, respectively. Extensive experiments validated that BridgeDrive yielded significant performance improvements in closed-loop planning tasks.

**Limitations and future work.** (1) Although the inference speed of BridgeDrive is suitable for real-time deployment, further acceleration can be achieved by distilling our diffusion model into a one-step planner without sacrificing the generation quality; see, e.g., Xie et al. (2024). (2) Despite BridgeDrive's extraordinary capacity to learn complex planning tasks, it still struggles to handle out-of-distribution scenarios, as illustrated in Appendix B.3. This limitation may be overcome by incorporating prior knowledge from VLA and post-training with reinforcement learning.

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

## A  RELATED WORK

**End-to-end autonomous driving.** Traditional motion planning pipelines often decompose the task into separate stages—perception, prediction, and planning—which inevitably introduce latency and information degradation across modules (Sadat et al., 2020). To overcome these limitations, recent studies have shifted toward planning-centric, end-to-end autonomous driving frameworks. End-to-end autonomous driving aims to map raw sensory inputs directly to trajectory predictions or control commands, enabling holistic system optimization that mitigates error propagation across modules (Wu et al., 2022; Zhang et al., 2021). UniAD (Hu et al., 2023) shows the feasibility of end-to-end autonomous driving by unifying multiple perception tasks to benefit planning. Building on this, VAD (Jiang et al., 2023) introduces compact vectorized scene representations to boost efficiency. VADv2 (Chen et al., 2024) proposes a probabilistic planning framework that models the distribution over possible actions and samples one for vehicle control. SimLingo (Renz et al., 2025) and GPTDriverV2 (Xu et al., 2025) incorporate vision-language understanding and language-action alignment, aiming to enhance closed-loop driving performance.

**Deterministic planners.** Some end-to-end autonomous driving planners relies on models such as multilayer perceptrons (MLPs) or variational autoencoders (VAEs). Transfuser (Chitta et al., 2023) and its extension Transfuser++ (Jaeger et al., 2023) exemplify this line of work by fusing multi-modal sensor inputs—such as camera images and LiDAR point clouds—through transformer-based encoders and decoding them into trajectory outputs via compact MLP heads. These models highlight the importance of effective sensor fusion in improving closed-loop driving performance. ORION (Fu et al., 2025) adopts a VAE-based latent planning architecture, which enables the model to capture multi-modal trajectory distributions while maintaining computational efficiency. These methods demonstrate how MLP and VAE-style architectures can serve as efficient baselines for end-to-end planning, though they often face limitations in modeling the full multi-modality of human driving behaviors compared to generative paradigms such as diffusion or flow-based models.

**Diffusion-based planners.** Diffusion policies provide a generative paradigm which can model the multi-modal nature of human driving behaviors with enhanced guidance control. Diffusion-ES (Yang et al., 2024) exhibits zero-shot instruction-following ability in planning. Diffusion-Planner (Zheng et al., 2025b) uses joint prediction modeling to achieve safe and adaptive planning. GoalFlow (Xing et al., 2025) leverages flow matching to produce diverse goal-conditioned trajectories and further uses a trajectory scorer to efficiently select trajectory using the goal point as a reference. DiffusionDrive (Liao et al., 2025) points out the issue of mode collapse, wherein the generated trajectories lack diversity, as different random noise inputs tend to converge to similar trajectories during the denoising process, proposing truncated diffusion policy that begins the denoising process from an anchored gaussian distribution instead of a standard Gaussian distribution to avoid mode collapse. TransDiffuser (Jiang et al., 2025) emphasizes another underlying bottleneck that leads to mode collapse in generated trajectories: The under-utilization of the encoded multi-modal conditional information. Therefore, it implements multi-modal representation decorrelation optimization mechanism during the denoising process, which aims to better exploit the multi-modal representation space to guide more diverse feasible planning trajectories from the continuous action space.

## B  ADDITIONAL RESULTS, VISUALIZATION AND LIMITATION

### B.1  COMPARISON WITH EXISTING WORKS

A comprehensive evaluation on Bench2Drive metrics is provided in Tables 5 and 6. Our method shows SOTA performance on both Driving Score (DS) and Success Rate (SR). Moreover, BridgeDrive exhibits outstanding multi-ability capability, especially in Merging (+11.17), and Traffic Sign (+7.02), resulting in overall improvement by +6.12 than SOTA. However, BridgeDrive demonstrates suboptimal performance in the Comfortness and Give Way metrics, suggesting a tendency toward frequent or poorly timed braking. This outcome may imply that our model prioritizes safety considerations, potentially at the expense of passenger comfort. This limitation should be addressed in the future work.

Table 5: Comprehensive comparison between BridgeDrive and baselines. BridgeDrive prioritizes safety over Comfortness.

| Method | Expert | Key technique | DS | SR(%) | Effi. | Comfort. |
|---|---|---|---|---|---|---|
| TCP-traj* | Think2Drive | CNN, MLP, GRU | 59.90 | 30.00 | 76.54 | 18.08 |
| UniAD-Base | Think2Drive | Transformer | 45.81 | 16.36 | 129.21 | 43.58 |
| VAD | Think2Drive | Transformer | 42.35 | 15.00 | 157.94 | 46.01 |
| DriveTransformer | Think2Drive | Transformer | 63.46 | 35.01 | 100.64 | 20.78 |
| ORION | Think2Drive | VLA+VAE | 77.74 | 54.62 | 151.48 | 17.38 |
| ORION diffusion | Think2Drive | VLA+Diffusion | 71.97 | 46.54 | N/A | N/A |
| DiffusionDrive[temp] | PDM-Lite | Diffusion | 77.68 | 52.72 | 248.18 | 24.56 |
| SimLingo | PDM-Lite | VLA | 85.07 | 67.27 | 259.23 | 33.67 |
| TransFuser++ | PDM-Lite | Transformer | 84.21 | 67.27 | N/A | N/A |
| BridgeDrive (ours) | PDM-Lite | Diffusion | 87.99 (+2.92) | 74.99 (+7.72) | 236.49 | 20.98 |

Table 6: Multi-ability evaluation results on Bench2Drive. BridgeDrive outperforms all baselines in all categories except for Give Way and Overtake.

| Method | Ability (%) | | | | | |
|---|---|---|---|---|---|---|
| | Merg. | Overtak. | Emer. Brake | Give Way | Traf. Sign | Mean |
| TCP-traj* | 8.89 | 24.29 | 51.67 | 40.00 | 46.28 | 34.22 |
| UniAD-Base | 14.10 | 17.78 | 21.67 | 10.00 | 14.21 | 15.55 |
| VAD | 8.11 | 24.44 | 18.64 | 20.00 | 19.15 | 18.07 |
| DriveTransformer | 17.57 | 35.00 | 48.36 | 40.00 | 52.10 | 38.60 |
| ORION | 25.00 | 71.11 | 78.33 | 30.00 | 69.15 | 54.72 |
| DiffusionDrive[temp] | 50.63 | 26.67 | 68.33 | 50.00 | 76.32 | 54.38 |
| SimLingo | 54.01 | 57.04 | 88.33 | 53.33 | 82.45 | 67.03 |
| TransFuser++ | 58.75 | 57.77 | 83.33 | 40.00 | 82.11 | 64.39 |
| BridgeDrive (ours) | 69.92 (+11.17) | 66.67 (-4.44) | 90.00 (+1.67) | 50.00 (-3.33) | 89.47 (+7.02) | 73.15 (+6.12) |

## B.2 INFERENCE SPEED

Inference wall-clock time comparison is detailed in Table 7. Note that the full diffusion model is slightly faster than BridgeDrive since it is approximately half the size of BridgeDrive. This is because full diffusion does not use anchors and thus omit all anchor-related cross-attention modules. BridgeDrive achieves reasonable inference speed even without any additional optimization, indicating its suitability for real-time deployment. It should be noted that in both DiffusionDrive and our BridgeDrive model, the primary computational cost stems from the perception and cross-attention modules. The diffusion process itself is applied only to a small trajectory matrix (8×2 or 11×2), contributing a minor portion of the total cost. Consequently, the overall computation time does not increase proportionally with the number of diffusion steps.

## B.3 CASE STUDY OF AN OUT-OF-DISTRIBUTION FAILURE SCENARIO

Despite the extraordinary modeling capacity of BridgeDrive, it cannot generalize to out-of-distribution scenarios, which is very common in closed-loop evaluation. For instance, the overtaking maneuver shall be aborted if oncoming vehicles are present in the adjacent lane. However, there are almost no such data in the training set. The reason is that training data are collected by a rule-based expert with privileged information (i.e., the expert has direct access to the ground truth of other traffic participants' location and dynamics). This expert has long-term planning capability and will only

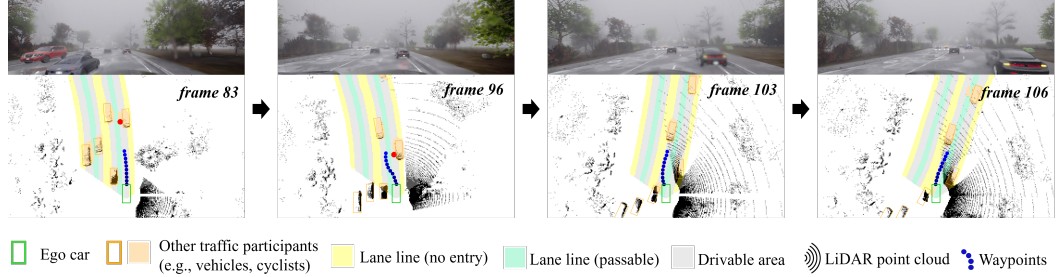

Figure 7: BridgeDrive cannot handle imperfect timing of lane-changing, which resulted from cumulative errors. This situation is outside of the training data distribution.

perform an overtaking maneuver when there is sufficient longitudinal space in adjacent lanes. Such an ideal timing is not always feasible in closed-loop evaluation due to cumulative difference between predicted and ground-truth speed. An example of imperfect timing for lane changing is provided in Fig. 7. This limitation may be overcome by integrating scene understanding prior knowledge from VLA into BridgeDrive or posting-training with reinforcement learning, which is left for future work.

## B.4 THE SUPERIORITY OF GEOMETRIC WAYPOINTS REMAINS UNDEREXPLORED

We briefly summarizes key experimental findings regarding temporal and geometric waypoints in autonomous driving models:

1. A prominent line of research has achieved SOTA performance primarily through the use of temporal waypoints (Liao et al., 2025; Zheng et al., 2025b; Chitta et al., 2023; Chen & Krähenbühl, 2022; Wu et al., 2022).

2. TransFuser++ (Jaeger et al., 2023) identified ambiguities inherent in temporal waypoints. To address this, the authors implemented a path predictor (analogous to our geometric waypoints) and a speed predictor. The speed is predicted via an MLP head using classification, which also outputs an associated uncertainty. Their experiments demonstrated that interpolating between target speeds, weighted by this uncertainty, effectively reduces collision rates.

3. SimLingo (Renz et al., 2025) trained a Vision-Language Model (VLM) to predict both temporal and geometric waypoints (termed "temporal speed waypoints" and "geometric path waypoints," respectively). They found that using only temporal waypoints resulted in poor steering performance, whereas incorporating geometric waypoints significantly improved vehicle control. In their framework, the control commands are derived from both representations: target speed is computed from temporal waypoints, while the steering angle is determined by geometric waypoints.

Table 7: Inference wall-clock time comparison. The full diffusion model is approximately half the size of BridgeDrive, as it does not have cross-attention modules for anchors. BridgeDrive achieves reasonable inference speed even without any additional optimization, indicating its suitability for real-time deployment.

| Configuration | Inference speed per frame | #diffusion timesteps |
|---|---|---|
| DiffusionDrive[temp] | 0.05 sec | 2 |
| DiffusionDrive[geo] | 0.05 sec | 2 |
| Full Diffusion[temp] | 0.05 sec | 100 |
| Full Diffusion[geo] | 0.05 sec | 100 |
| BridgeDrive[temp] (ours) | 0.10 sec | 20 |
| BridgeDrive[geo] (ours) | 0.10 sec | 20 |

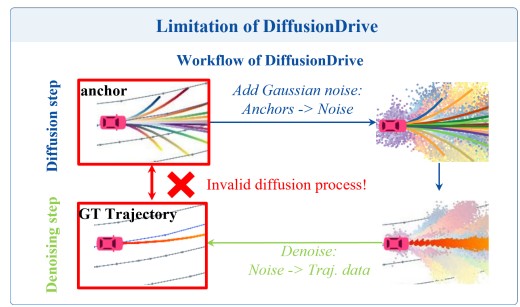 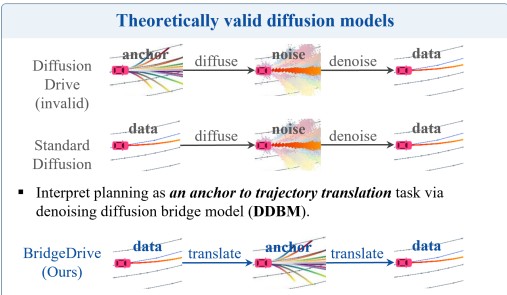

Figure 8: The starting point of the forward process (anchor) should match the endpoint of the reverse process (ground truth trajectory), but they are inconsistent in DiffusionDrive. Figures are adapted from Liao et al. (2025)

4. DriveGPT4-V2 (Xu et al., 2025) does not directly output waypoints for control. Instead, it predicts a target speed and steering angle directly, using temporal and geometric waypoints (referred to as "waypoints" and "route points") solely as supervisory signals during training. Their ablation study concluded that predicting the final control commands (speed and angle) is more effective than predicting intermediate waypoints.

These findings suggest that optimal control performance may not be achieved by relying exclusively on either temporal or geometric waypoints. Instead, superior performance likely arises from the interplay between these two representations and their derived control variables. The field has not yet reached a definitive conclusion on this matter.

A comprehensive understanding of the fundamental roles of temporal and geometric waypoints would require an extensive experimental evaluation of various SOTA algorithms across multiple mainstream benchmarks, which is left for future work. Therefore, in our experiments, we adopt the conventional practice established by TransFuser++ (Jaeger et al., 2023), as our model architecture most closely resembles theirs. Consequently, the experiments presented in this paper primarily demonstrate the superiority of geometric waypoints within the Bench2Drive benchmark.

### B.5 CONCEPTUAL COMPARISON BETWEEN A STANDARD FULL DIFFUSION MODEL, DIFFUSIONDRIVE, AND BRIDGEDRIVE

BridgeDrive is distinguished by its theoretical rigor, as it restores the inherent symmetry of diffusion models, addressing a fundamental flaw in prior truncated diffusion approaches. An illustration is also visualized in Fig. 8.

- **Standard full diffusion model:**
  `ground truth trajectory → (forward diffusion) Gaussian noise → (reverse denoising) ground truth trajectory`
  ✓ *Diffusion symmetry preserved!*

- **DiffusionDrive:**
  `anchor → (forward diffusion) noised anchor → (reverse denoising) ground truth trajectory`
  ✗ *Diffusion symmetry violated!*
  *Issue: The starting point of the forward process (anchor) should match the endpoint of the reverse process (ground truth trajectory), but they are inconsistent.*

- **BridgeDrive (Ours):**
  `ground truth trajectory → (forward diffusion bridge) anchor → (reverse denoising bridge) ground truth trajectory`
  ✓ *Diffusion symmetry preserved!*

Table 8: Comparison among the major modules in the model architectures of DiffusionDrive, DiffusionDrive[temp], DiffusionDrive[geo], and BridgeDrive.

| Module | DiffusionDrive | DiffusionDrive[temp] | DiffusionDrive[geo] | BridgeDrive |
|---|---|---|---|---|
| Perception | Transfuser | Transfuser++ | Transfuser++ | Transfuser++ |
| Classifier | Transfuser | Transfuser++ | Transfuser++ | BridgeDrive |
| Denoiser | DiffusionDrive | DiffusionDrive | DiffusionDrive | BridgeDrive |
| Output | Temporal waypoints | Temporal waypoints | Geometric waypoints | Geometric waypoints |

## C EXPERIMENT DETAILS

### C.1 DATASETS FILTERING AND AUGMENTATION

Zimmerlin et al. (2024) proposed a data filtering method to reduce redundancy in training datasets. The method involves keeping frames where significant changes occur compared to the previous frame. Specifically, a frame is retained if either of the following conditions is met.

- The target speed changes by more than $0.1$ m/s.

- The angle to any predicted geometric path waypoints changes by more than $0.5°$.

From the remaining frames, 14% are randomly selected and kept. This strategy results in a 50% reduction in the dataset size.

Additionally, the authors adjust the expert's driving style by modifying behaviors, such as the timing of braking when approaching pedestrians. This adjustment ensures the expert's actions are more interpretable and provide clearer learning signals for the model. Furthermore, the paper removes class weights for target speed values, particularly for over-represented classes like braking, to avoid biasing the model towards more frequent behaviors. This ensures the model learns from frames critical for driving tasks, rather than focusing on frequent but less important ones.

### C.2 ADAPTATION OF DIFFUIONDRIVE TO BENCH2DRIVE BENCHMARK

We denote adapted versions of DiffusionDrive as DiffusionDrive[temp] and DiffusionDrive[geo]. We explain our adaption from four aspects.

**Perception module.** The original DiffusionDrive was built on the backbone of Transfuser (Chitta et al., 2023), whereas our BridgeDrive is based on Transfuser++ (Jaeger et al., 2023). To ensure a competitive baseline for fair comparison, for the adaptation of DiffusionDrive, we use the perception module from Transfuser++, which is proven to achieve SOTA on Bench2Drive benchmark (Zimmerlin et al., 2024).

**Denoiser.** We keep the model architecture of the denoiser identical to its original design as it is unique to each model under comparison.

**Classifier.** The classifier in DiffusionDrive consists of cross-attention modules to process the perception features. We keep its architecture in line with the perception module.

**Output.** The output trajectory representation of DiffusionDrive[temp] is temporal waypoints, which is kept the same as DiffusionDrive. The analysis of the impact of output representation is provided in Section 4.2, where DiffusionDrive[temp]'s geometric waypoints counterpart DiffusionDrive[geo] is implemented and evaluated.

An overview comparing the architectures of the major modules across DiffusionDrive, DiffusionDrive[temp], and BridgeDrive is provided in Table 8. The rest of the implementation and training details of DiffusionDrive[temp] are kept the same as BridgeDrive for a fair comparison.

Table 9: Influence of anchor classification accuracy on the performance of BridgeDrive.

| Anchor-selected | Best | 2nd best | 3rd best | 4th best |
|---|---|---|---|---|
| Success rate (%) | 74.99 | 69.09 | 61.36 | 57.72 |
| Driving score | 87.99 | 85.31 | 80.76 | 77.53 |

## C.3 IMPLEMENTATION AND TRAINING DETAILS

As mentioned in Section 3.4, our model consists of three modules. For perception module, we keep the original design as described in (Jaeger et al., 2023) and (Zimmerlin et al., 2024). Once the perception module is pre-trained, it is frozen during the training phase of the denoiser and classifier modules. The joint loss for the denoiser and classifier is defined as:

$$L_{\text{overall}} = w_{\text{diffusion}} L_{\text{diffusion}} + w_{\text{classification}} L_{\text{classification}}, \tag{11}$$

where $L_{\text{diffusion}}$ is as defined in Eq. (9) and $L_{\text{classification}}$ is the cross-entropy loss. However, in order to ensure a fair comparison with Liao et al. (2025), we replace the MSE loss in $L_{\text{diffusion}}$ with L1 loss. By default, both $w_{\text{diffusion}}$ and $w_{\text{classification}}$ are set to 1. We optimize them using AdamW (Loshchilov & Hutter, 2017a) with a cosine annealing learning schedule (Loshchilov & Hutter, 2017b). The learning rate is set as $\text{lr}_0 = 3 \times 10^{-4}$, $T_0 = 10$, $T_{mult} = 2$. We vary the number $N_{\text{anchor}}$ of anchors in BridgeDrive and find that $N_{\text{anchor}} = 60$ is optimal. Our models are trained for 10 epochs on a single NVIDIA H20 GPU, which takes around 10 hours.

For diffusion schedule, we employ the variance preserving (VP) schedule from Karras et al. (2022). Specifically, we first define $s(t) = 1/\sqrt{e^{\beta_d t^2/2 + \beta_{min} t}}$ and $\sigma(t) = \sqrt{e^{\beta_d t^2/2 + \beta_{min} t} - 1}$. We then set the diffusion coefficients in the forward diffusion bridge SDE Eq. (5) to $f(t) = \dot{s}(t)/s(t)$ and $g(t) = \sqrt{2s(t)^2 \dot{\sigma}(t)\sigma(t)}$. We choose $\beta_d = 2.0$ and $\beta_{min} = 0.1$ following Zheng et al. (2025a).

## D ADDITIONAL ABLATION STUDY

### D.1 INFLUENCE OF ANCHOR CLASSIFICATION ACCURACY

To assess the impact of anchor selection, we generated trajectories using sub-optimal anchors (i.e., the 2nd, 3rd, and 4th most likely from the classifier). The result is presented in Table 9. Our bridge diffusion model exhibited significant resilience, achieving $> 60\%$ success rate with the 2nd and 3rd anchors. However, both success rate and driving score decreased as lower-probability anchors were chosen, which demonstrate the importance of anchor classification accuracy.

### D.2 INFLUENCE OF DIFFUSION BRIDGE MODULE AND ANCHOR PRIOR

We perform ablation study to quantify the contribution of diffusion bridge module and anchor, respectively.

To isolate the contribution of the diffusion blocks, we construct a BridgeDrive model with only 1 anchor. As shown in Table 10, without the prior information of the anchor, the BridgeDrive model achieves a performance comparable to that of the full diffusion model in Table 2.

To isolate the contribution of the diffusion blocks, we conducted an ablation where we used only the anchor selector (without any diffusion refinement) on the Bench2Drive benchmark. As shown in Table 10, the anchor-only model fails to achieve competent performance, even with a very large number of anchors. This provides compelling evidence that diffusion blocks are essential for generating high-quality trajectories and are not merely minor enhancement.

In addition, we constructed a regression model by removing the time-embedding component from our denoiser model while keeping the rest of the architecture unchanged. Table 10 shows that the anchor regression model performs consistently worse than our BridgeDrive model. This supports our claim that the iterative, probabilistic refinement provided by the diffusion bridge process in BridgeDrive is essential for achieving higher performance.

Table 10: Influence of the number of anchors on the performance of BridgeDrive and anchor-based classification and regression planning models.

| Number of anchors | $k=1$ | $k=20$ | $k=40$ | $k=60$ | $k=80$ | $k=200$ | $k=500$ | $k=1000$ |
|---|---|---|---|---|---|---|---|---|
| **Success rate (%)** | | | | | | | | |
| Anchor classification | - | 2.72 | 8.18 | 16.81 | 19.54 | 25.92 | 36.81 | 36.36 |
| Anchor-based regression | - | 68.18 | 72.72 | 70.91 | 70.91 | - | - | - |
| BridgeDrive (ours) | 67.27 | 72.27 | 73.18 | 74.99 | 72.72 | - | - | - |
| **Driving score** | | | | | | | | |
| Anchor classification | - | 27.71 | 35.78 | 45.43 | 49.02 | 57.89 | 63.12 | 62.3 |
| Anchor-based regression | - | 86.73 | 87.09 | 86.91 | 86.77 | - | - | - |
| BridgeDrive (ours) | 84.88 | 87.02 | 87.24 | 87.99 | 87.27 | - | - | - |

## D.3 Influence of the Number of Anchors

The impact of the number of anchors, which directly influences anchor diversity, was evaluated through an ablation study. The results in Table 10 indicate that BridgeDrive's success rate initially rises with the number of anchors, peaking at 60 and affirming the positive role of diversity. A subsequent decline in performance suggests that a larger anchor set compromises classification accuracy. Consequently, the model's optimal performance is achieved at an equilibrium between anchor diversity and classification precision.

## E Preliminary Results on LEAD Dataset

LEAD (Nguyen et al., 2026), the latest advancement in end-to-end autonomous driving, thoroughly investigates the generalization gap between expert and student policies, identifies and analyzes key sources of misalignment that impede effective imitation in CARLA, and uses these insights to produce a novel expert policy and corresponding dataset explicitly designed to mitigate Learner–Expert Asymmetry in driving.

This section presents a preliminary experiment evaluating BridgeDrive on the LEAD datasets. The latest progress will be made available at https://github.com/shuliu-ethz/BridgeDrive. The configuration is detailed as follows and differs slightly from previous experiments:

1. Speed is removed from the anchor clustering process; the resulting anchors encode only steering information. A total of 60 anchors are used in this adaptation.

2. During training, the LEAD perception module is reused and kept frozen, while the denoiser and classifier are trained for 30 epochs.

3. At inference time, BridgeDrive outputs only geometric waypoints for steering maneuvers, with speed controlled by the LEAD outputs.

Key results are presented in Table 11 (also shown in the main text) and Table 12. BridgeDrive achieves performance comparable to LEAD. Notably, its success rate and driving score are $2.45\%$ and $1.14$ higher than that of LEAD. The evaluation indicates that BridgeDrive generalizes well across different training sets. Further improvements are expected through a more thorough investigation of anchor quantity, diffusion parameters, learning rate, training duration, and the speed control mechanism.

## F Declaration

We used large language models (LLMs) to polish writing.

Table 11: Comparison between BridgeDrive and LEAD.

| Method | Expert | DS | SR(%) | Effi. | Comfort. |
|---|---|---|---|---|---|
| TFv6 (Nguyen et al., 2026) | LEAD | 95.2 $\pm$ 0.3 | 86.8 $\pm$ 0.7 | N/A | N/A |
| BridgeDrive (ours) | PDM-Lite | 87.99 $\pm$ 0.67 | 74.99 $\pm$ 1.35 | **236.49** $\pm$ 2.32 | 20.98 $\pm$ 0.74 |
| BridgeDrive (ours) | LEAD | **96.34** $\pm$ 0.55 | **89.25** $\pm$ 0.50 | 202.92 $\pm$ 3.27 | **23.24** $\pm$ 1.06 |

Table 12: Multi-ability evaluation results of BridgeDrive on PDM-Lite and LEAD datasets.

| Method | Ability (%) | | | | | |
|---|---|---|---|---|---|---|
| | Merg. | Overtak. | Emer. Brake | Give Way | Traf. Sign | Mean |
| BridgeDrive (PDM-Lite) | 69.92 | 66.67 | 90.00 | 50.00 | 89.47 | 73.15 |
| BridgeDrive (LEAD) | 76.25 | 95.56 | 96.67 | 50.00 | 92.63 | 82.22 |

