# OpenReview forum: "BridgeDrive: Diffusion Bridge Policy for Closed-Loop Trajectory Planning in Autonomous Driving"
_ICLR.cc/2026/Conference — ICLR 2026 Poster_

### Official Review · Reviewer_Awhw · 2025-10-24

**Soundness:** 3
**Presentation:** 3
**Contribution:** 3
**Rating:** 6
**Confidence:** 4

**Summary:**

This paper proposes BridgeDrive, a planning method based on diffusion bridges that addresses the theoretical inconsistency in prior anchor-based diffusion policies such as DiffusionDrive. By enforcing a symmetric and theoretically consistent forward–reverse process, BridgeDrive improves closed-loop performance. The paper further investigates temporal speed waypoints vs. geometric path waypoints representations and demonstrates that the geometric representation leads to less likely to violate route lane constrain. Experiments on Bench2Drive show clear improvements over prior state of the art.

**Strengths:**

1. By incorporating bridge diffusion, the paper builds an explicit and theoretically complete formulation that connects anchor information with trajectory generation in anchor-based diffusion policies. This directly addresses the theoretical shortcomings arising when truncated diffusion processes ignore boundary consistency and rely on heuristically truncating the diffusion process.
2. The comparison between temporal and geometric trajectory parameterizations provides valuable insight for practical control and shows the superiority of geometric spacing.
3. The method achieves notable improvements on Bench2Drive, demonstrating strong closed-loop performance benefits.
4. Real-time inference and robust multi-modal trajectory generation contribute to the practicality of the approach.

**Weaknesses:**

1. The training and evaluation are limited to Bench2Drive simulation. Validation on larger-scale real human driving datasets, e.g. NavSim,  would be crucial to support claims of real-world generalization.

2. The influence of scene context versus anchor selection on the final trajectory is insufficiently analyzed. More evidence is needed to show that the trajectory is shaped jointly by scene understanding and anchors, instead of the anchor having disproportionate dominance.

**Questions:**

1. Could you provide quantitative or qualitative results confirming that both scene features and anchors significantly contribute to the trajectory generation (for example, attribution studies or case analyses)?

2. How do the number, diversity, and classification accuracy of anchors impact planning robustness? Any sensitivity or failure-case analysis?

3. Are there any experiments on large-scale datasets such as nuPlan or NavSim to support broader applicability claims?

---

> ### Author Response · Authors · 2025-11-25
> **Response to Reviewer Awhw (1/3)**
>
> Thank you for your valuable comments! Below, we address your concerns point by point.
>
> > Weakness 1 and Question 3: Validation on larger-scale real human driving datasets, e.g. NavSim, would be crucial to support claims of real-world generalization.
>
>
> Thank you for the suggestion! We conducted additional experiments for our method on the **NAVSIM** dataset. Following the setup of DiffusionDrive [1], we evaluate our model using the PDM score (PDMS), a comprehensive metric integrating no at-fault collisions (NC), drivable area compliance (DAC), time-to-collision (TTC), comfort (Comf.), and ego progress (EP).
>
>
> | Method | NC | DAC | TTC | Comf. | EP | PDMS |
> | -------- | -------- | -------- |-------- |-------- |-------- |-------- |
> | **DiffusionDrive (reported in [1])** | 98.2 | 96.2 | 94.7 | 100 | 82.2| 88.1 |
> | **DiffusionDrive (reproduced with different random seed)** | 98.2  | 95.9 | 94.3 | 100 | 81.9 | 87.6 |
> | **BridgeDrive (ours)** | 98.2 | 96.1 | 94.5 | 100 | 82.3 | 88.0 |
>
> The results in the above table demonstrate that our BridgeDrive method attains competitive performance on the NAVSIM benchmark. We posit that this performance is notable, as NAVSIM is a mature dataset, where existing state-of-the-art models have already **saturated** its performance potential (NAVSIM is after all a psuedo closed-loop dataset with **non-reactive agents** under a strong assumption of **NO** environment and agent interaction over a short horizon).
>
> For this reason, our paper primarily targets the **more realistic closed-loop setting** (i.e., Bench2Drive), as emphasized in the paper title. The closed-loop setting is more challenging and better reflects a policy’s real-world planning capability, since the ego vehicle’s decisions influence both its own future states and those of surrounding agents. This interaction creates a feedback loop that can amplify small prediction errors over time, as discussed in Section 2.1. We also highlight that evaluating in closed-loop settings has become an emerging trend in the autonomous driving community [2,3,4,5], driving more realistic assessments that better reflect actual technological progress. Our experimental results demonstrated that our BridgeDrive method outperformed previous state-of-the-art methods in this challenging setting.
>
> [1] Liao, et al. DiffusionDrive: Truncated Diffusion Model for End-to-End Autonomous Driving. CVPR 2025.
>
> [2] Caesar, et al. nuplan: A closed-loop ml-based planning benchmark for autonomous vehicles. arXiv:2106.11810, 2021.
>
> [3] Chitta, et al. Transfuser: Imitation with transformer-based sensor fusion for autonomous driving. TPAMI 2022.
>
> [4] Zheng, et al. Diffusion-based planning for autonomous driving with flexible guidance. ICLR 2025.
>
> [5] Yang, et al. DriveArena: A Closed-loop Generative Simulation Platform for Autonomous Driving. ICCV 2025.

---

> ### Author Response · Authors · 2025-11-25
> **Response to Reviewer Awhw (2/3)**
>
> > Weakness 2 & Question 1: More evidence is needed to show that the trajectory is shaped jointly by scene understanding and anchors, instead of the anchor having disproportionate dominance. Could you provide quantitative or qualitative results confirming that both scene features and anchors significantly contribute to the trajectory generation (for example, attribution studies or case analyses)?
>
> Thank you for the insightful comments!
>
> We perform ablation study to quantify the contribution of bridge diffusion module and anchor, respectively.
>
> To isolate the contribution of anchors, we construct a BridgeDrive model with only 1 anchor. As shown in the table below, without the prior information of the anchor, the BridgeDrive model achieves a comparable performance as full diffusion model (as described in Table 2 of our paper).
>
> To isolate the contribution of the diffusion block, we conducted an ablation where we used only the anchor selector (without any diffusion refinement) on the Bench2Drive benchmark. As shown in the following table, the anchor-only model fails to achieve competent performance, even with a very large number of anchors. This provides compelling evidence that the diffusion blocks are essential for generating high-quality trajectories and are not merely a minor enhancement.
>
> | **Metric / Number of Anchors** | **k=1** | **k=20** | **k=40** | **k=60** | **k=80** | **k=200** | **k=500** | **k=1000** |
> |---------------------------|----------|----------|----------|----------|-----------|-----------|------------|-----|
> | **Success Rate (%)**      | |            |          |          |          |           |           |            |
> |   BridgeDrive (Ours)     | 67.27     | 72.27    | 73.18    | **74.99**| 72.72    | —         | —         | —          |
> |   Anchor-Only (Ablation) | —     | 2.72     | 8.18     | 16.81    | 19.54    | 25.92     | 36.81     | 36.36      |
> | **Driving Score**          |  |          |          |          |          |           |           |            |
> |   BridgeDrive (Ours)      | 84.88    | 87.02    | 87.24    | **87.99**| 87.27    | —         | —         | —          |
> |   Anchor-Only (Ablation) | —    | 27.71    | 35.78    | 45.43    | 49.02    | 57.89     | 63.12     | 62.30      |
>
>
> A qualitative illustration is given in Figure 1 of our paper. As the red trajectory illustrated, given the same traffic scene, a catastrophically wrong anchor diffuses to a wrong trajectory. In comparison, the blue one selected the correct anchor, which diffuses to a valid trajectory under the same guidance of the traffic scene.
>
> **In summary,** we have demonstrated that:
> 1.  Removing the diffusion component causes a catastrophic drop in performance, proving its necessity.
> 2.  The prior information provided by anchor can further improve trajectory planning quality.

---

> ### Author Response · Authors · 2025-11-25
> **Response to Reviewer Awhw (3/3)**
>
> > Question 2: How do the number, diversity, and classification accuracy of anchors impact planning robustness? Any sensitivity or failure-case analysis?
>
> Thank you for this very thoughtful sugguestion, which helps us greatly improve the comprehensiveness of this paper.
>
> First, we investigate the impact of classification accuracy of anchors. We perform experiments, by choosing 2nd, 3rd, and 4th most likely anchor according to our anchor classifier, and generate vehicle trajectory from these sub-optimal anchors. The experiment showed that, our bridge diffusion model demonstrated certain resiliance against wrong anchor and still a success rate above 60% when 2nd or 3rd most likely anchors are selected. However, we can also observe that the success rate drops as less likely anchors are selected.
>
> | anchor-selected | 1st | 2nd | 3rd | 4th |
> | -------- | -------- | -------- |-------- |-------- |
> | **Success Rate (%)**     | 74.99  | 69.09    | 61.36 | 57.72 |
> | **Driving Score**     | 87.99  | 85.31    | 80.76 | 77.53 |
>
>
> Second, we performed an ablation on the number of anchors, which also reflect the influence of anchor diversity. As shown in the table, the success rate of BridgeDrive rises as number of anchors increases from 20 to 60, demonstrating the positive contribution of anchor diversity. However, as the number of anchor further increases, the success rate starts to drop. The reason could be attributed to reduced classification accuracy due to increase number of anchors. The optimal performance of BridgeDrive is achieved via the balance between anchor diversity and anchor classification accuracy.
>
> | **Metric / Number of Anchors** | **k=20** | **k=40** | **k=60** | **k=80** |
> |---------------------------|----------|----------|----------|----------|
> |   **Success Rate (%)**      | 72.27    | 73.18    | **74.99**| 72.72    |
> |   **Driving Score**      | 87.02    | 87.24    | **87.99**| 87.27    |
>
> Third, an example of failure-case analysis is given in Figure 1. As the red trajectory illustrated, given the same traffic scene, a catastrophically wrong anchor diffuses to a wrong trajectory. In comparison, the blue one selected the correct anchor, which diffuses to a valid trajectory under the same guidance of the traffic scene. We rewrite and highlight this part of the paper to make it more clear to readers.
>
> Thank you again for your thoughtful comments, which have greatly helped us improve the clarity and quality of our paper. We hope that we have adequately addressed all of your concerns. Please let us know if you have any further questions or concerns.

---

> > ### Comment · Reviewer_Awhw · 2025-11-25
> >
> > Thanks the author for the rigorous and comprehensive supplementary experiments and patient replies. BridgeDrive truly provides a complete and empirical theoretical framework for combining anchor-based methods with diffusion.
> >
> > Now I am satisfied with the paper and have raised my score.

---

### Official Review · Reviewer_RMqE · 2025-10-28

**Soundness:** 2
**Presentation:** 3
**Contribution:** 2
**Rating:** 4
**Confidence:** 4

**Summary:**

This paper introduces BridgeDrive, which adapts the Denoising Diffusion Bridge Model to generate geometric path waypoints from an anchor distribution, achieving state-of-the-art performance on the widely used Bench2Drive benchmark.

**Strengths:**

1. The paper point out that the problem of DiffusionDrive is that it is not a theoretical diffusion process. This issue has misled the community. The authors identified this and provided a solution.
2. Using Bridge Diffusion to solve this makes sense.

**Weaknesses:**

1. Given the authors' claim that geometric waypoints outperform temporal waypoints, it is better to validate whether this indicates a bias in the Bench2Drive benchmark. Therefore, verification across more benchmarks is highly recommended.

2. The architecture proposed in Figure 2 lacks ablation studies on its structural design.

3. The use of anchors greatly simplifies the true trajectory distribution, raising the question of whether using diffusion is truly necessary. A compelling comparison with SOTA anchor-based methods is needed to justify the adoption of bridge diffusion.

**Questions:**

1. Why is the Think2Drive expert used as a baseline when diffusion-based methods appear to perform worse than other approaches? Using the same expert for all methods facilitates a fair comparison.

2. Is Bridge Diffusion more challenging to train than standard diffusion models, and does it still support both classifier-free guidance and classifier guidance?

---

> ### Author Response · Authors · 2025-11-25
> **Response to Reviewer RMqE (1/3)**
>
> Thank you for your valuable comments! Below, we address your concerns point by point.
>
> > Weakness 1: Given the authors' claim that geometric waypoints outperform temporal waypoints, it is better to validate whether this indicates a bias in the Bench2Drive benchmark...
>
> Thank you for pointing out this important issue! After thorough literature review, we briefly summarize the key experimental findings regarding temporal and geometric waypoints in autonomous driving models below.
>
> 1. A prominent line of research has achieved state-of-the-art (SOTA) performance primarily through the use of temporal waypoints [1-5].
>
> 2. TransFuser++ [6] identified ambiguities inherent in temporal waypoints. To address this, the authors implemented a path predictor (analogous to our geometric waypoints) and a speed predictor. The speed is predicted via an MLP head using classification, which also outputs an associated uncertainty. Their experiments demonstrated that interpolating between target speeds, weighted by this uncertainty, effectively reduces collision rates.
>
> 3. SimLingo [7] trained a Vision-Language Model (VLM) to predict both temporal and geometric waypoints (termed "temporal speed waypoints" and "geometric path waypoints," respectively). They found that using only temporal waypoints resulted in poor steering performance, whereas incorporating geometric waypoints significantly improved vehicle control. In their framework, the control commands are derived from both representations: target speed is computed from temporal waypoints, while the steering angle is determined by geometric waypoints.
>
> 4. DriveGPT4-V2 [8] does not directly output waypoints for control. Instead, it predicts a target speed and steering angle directly, using temporal and geometric waypoints (referred to as "waypoints" and "route points") solely as supervisory signals during training. Their ablation study concluded that predicting the final control commands (speed and angle) is more effective than predicting intermediate waypoints.
>
> These findings suggest that optimal control performance may not be achieved by relying exclusively on either temporal or geometric waypoints. Instead, superior performance likely arises from the interplay between these two representations and their derived control variables. The field has not yet reached a definitive conclusion on this matter.
>
> A comprehensive understanding of the fundamental roles of temporal and geometric waypoints would require an extensive experimental evaluation of various SOTA algorithms across multiple mainstream benchmarks, which is beyond the scope of this paper. Therefore, in our experiments, we adopted the conventional practice established by TransFuser++ [6], as our model architecture most closely resembles theirs. Consequently, the experiments presented in this paper primarily demonstrate the superiority of geometric waypoints within the Bench2Drive benchmark as shown in Table 2. We explicitly acknowledged this limitation in the relevant sections of the paper.
>
> [1] Liao et al. DiffusionDrive: Truncated diffusion model for end-to-end autonomous driving. CVPR 2025.
>
> [2] Chen, et al. Learning from all vehicles. CVPR 2022.
>
> [3] Chitta, et al. Neat: Neural attention fields for end-to-end autonomous driving. ICCV 2021.
>
> [4] Wu, et al. Trajectory-guided control prediction for end-to-end autonomous driving: A simple yet strong baseline. NeurIPS 2022.
>
> [5]Zheng, et al. Diffusion-based planning for autonomous driving with flexible guidance. ICLR 2025
>
> [6] Jaeger, et al. Hidden biases of end-to-end driving models. ICCV 2023.
>
> [7] Renz, et al. Simlingo: Vision-only closed-loop autonomous driving with language-action alignment. CVPR 2025.
>
> [8] Xu, et al. Drivegpt4-v2: Harnessing large language model capabilities for enhanced closed-loop autonomous driving. CVPR 2025
>
> > Weakness 2: The architecture proposed in Figure 2 lacks ablation studies on its structural design.
>
> Since the main motivation of BridgeDrive is to address the weaknesses of DiffusionDrive, the architecture shown in Figure 2 is directly adopted from DiffusionDrive to ensure a fair comparison between the two methods. Additionally, the Transfuser-series [1,2] and DiffusionDrive [3] stated that this architecture was extensively tuned.
>
> [1] Chitta, et al. Transfuser: Imitation with transformer-based sensor fusion for autonomous driving. TPAMI 2022.
>
> [2] Jaeger, et al. Hidden biases of end-to-end driving models. ICCV 2023.
>
> [3] Liao, et al. DiffusionDrive: Truncated diffusion model for end-to-end autonomous driving. CVPR 2025.

---

> ### Author Response · Authors · 2025-11-25
> **Response to Reviewer RMqE (2/3)**
>
> > Weakness 3: Is diffusion truly necessary? A compelling comparison with SOTA anchor-based methods is needed to justify the adoption of bridge diffusion.
>
>
> Thank you for this insightful comment! We strongly believe the diffusion blocks are a crucial component of our design, and we provide two key empirical justifications below.
>
> **1. Superior Performance Over SOTA Anchor-Based Methods.**
> Anchor-based methods represent a common and strong baseline in autonomous driving. For a rigorous comparison, we benchmark our method against two recent, high-performing anchor-based methods: VADv2 [1] and Hydra-MDP [2]. _As these are preprint works without public code_, we compare performance on the __NAVSIM__ benchmark using the results reported in their respective papers. The table below demonstrates that our method outperforms these SOTA anchor-based approaches, underscoring the additional performance gain provided by the diffusion module.
>
>
> | Method | NC | DAC | TTC | Comf. | EP | PDMS |
> | -------- | -------- | -------- |-------- |-------- |-------- |-------- |
> | VADv2-V8192 [1] | 97.2 | 89.1 | 91.6 | 100 | 76.0| 80.9 |
> | Hydra-MDP-V8192-W-EP [2] | 98.3 | 96.0 | 94.6 | 100 | 78.7| 86.5 |
> | BridgeDrive (Ours) | 98.2 | 96.1 | 94.5 | 100 | 82.3 | **88.0** |
>
> **2. Ablation Study: Anchor Selector Alone Fails Catastrophically.**
> To further isolate the contribution of the diffusion blocks, we conducted an ablation where we used only the anchor selector (without any diffusion refinement) on the Bench2Drive benchmark. As shown in the following table, the anchor-only model fails to achieve competent performance, even with a very large number of anchors. This provides compelling evidence that the diffusion blocks are essential for generating high-quality trajectories and are not merely a minor enhancement.
>
> | **Metric / Number of Anchors** | **k=20** | **k=40** | **k=60** | **k=80** | **k=200** | **k=500** | **k=1000** |
> |---------------------------|----------|----------|----------|----------|-----------|-----------|------------|
> | **Success Rate (%)**      |          |          |          |          |           |           |            |
> |   BridgeDrive (Ours)      | 72.27    | 73.18    | **74.99**| 72.72    | —         | —         | —          |
> |   Anchor-Only (Ablation)  | 2.72     | 8.18     | 16.81    | 19.54    | 25.92     | 36.81     | 36.36      |
> | **Driving Score**         |          |          |          |          |           |           |            |
> |   BridgeDrive (Ours)      | 87.02    | 87.24    | **87.99**| 87.27    | —         | —         | —          |
> |   Anchor-Only (Ablation)  | 27.71    | 35.78    | 45.43    | 49.02    | 57.89     | 63.12     | 62.30      |
>
> **In summary,** we have demonstrated that:
> 1.  Our BridgeDrive model surpasses the performance of SOTA anchor-based methods.
> 2.  Removing the diffusion component causes a catastrophic drop in performance, proving its necessity.
>
> Coupled with our paper's discussion that a pure diffusion approach also underperforms, these results robustly validate the importance of our hybrid design, which successfully combines the strengths of both anchor-based and diffusion paradigms.
>
>
> [1] Chen, et al. Vadv2: End-to-end vectorized autonomous driving via probabilistic planning. arXiv:2402.13243, 2024.
>
> [2] Li, et al. Hydra-MDP: End-to-end multimodal planning with multi-target hydra-distillation. arXiv:2406.06978, 2024.

---

> ### Author Response · Authors · 2025-11-25
> **Response to Reviewer RMqE (3/3)**
>
> > Question 1: Why is the Think2Drive expert used as a baseline when diffusion-based methods appear to perform worse than other approaches? Using the same expert for all methods facilitates a fair comparison.
>
> This is mainly a result of specific designs of baseline methods. For example, DriveTransformer [1] has a sensor cross attention to enhance BEV feature from surrounding camera input (6 images). Likewise, many E2E methods consider taking advantage of multi-view inputs. Meanwhile, Think2Drive as expert collects data from Carla simulation with surrounding views, which is compatible with some other real-world datasets, e.g., nuScenes. Hence, we do not change the setup of these baselines. Meanwhile, our focus is on diffusion-based planner, which replies on single-camera input of PDM-Lite [2]. The result shows that even with single-camera input (with Lidar), our method achieves SOTA.
>
> [1] Jia, et al. DriveTransformer: Unified transformer for scalable end-to-end autonomous driving. ICLR 2025.
>
> [2] Liao, et al. DiffusionDrive: Truncated diffusion model for end-to-end autonomous driving. CVPR 2025.
>
>
> > Question 2: Is Bridge Diffusion more challenging to train than standard diffusion models, and does it still support both classifier-free guidance and classifier guidance?
>
> Thank you for this great question! The difficulty of training bridge diffusion models is the **same** as that of training standard diffusion models, since both optimize L2 regression losses, as shown in Equations 3 and 9. Empirically, we also observe that the convergence speed of diffusion bridge is similar to that of standard diffusion models.
>
> Yes, diffusion bridge still supports both classifier-free guidance and classifier guidance. For example, classifier-free guidance for diffusion bridge is given by $x_\theta^{CFG}(x_t,t,y,z) = w * x_\theta(x_t,t,y,\emptyset) + (1-w) * x_\theta(x_t,t,y,z)$.
>
>
> Thank you again for your valuable comments, which have greatly helped us improve the clarity and quality of our paper. We hope that we have adequately addressed all of your concerns. Please let us know if you have any further questions or concerns.

---

> > ### Comment · Reviewer_RMqE · 2025-11-27
> >
> > Thank you for the detailed reply. I still have a few questions:
> > 1. Does the NavSim experiment use temporal trajectories?
> > 2. In the section "Ablation Study: Anchor Selector Alone Fails Catastrophically," directly using anchors would definitely not be effective. Therefore, the experimental comparison here is not particularly reasonable. A more meaningful approach would be to compare refining anchors using a regression model versus refining them with diffusion, as this would better illustrate the point.
> > 3. On Navsim, the scores for BridgeDrive and DiffusionDrive are similar. Could you try analyzing why?

---

> ### Author Response · Authors · 2025-11-28
> **Response to Reviewer RMqE's further questions**
>
> Thank you for your reply and for the follow-up questions!
>
> > 1. Does the NavSim experiment use temporal trajectories?
>
> Yes. Both our BridgeDrive method and DiffusionDrive use temporal trajectories on NAVSIM. For fairness, we strictly follow the experimental setup of DiffusionDrive to ensure a fair comparison on NAVSIM.
>
> > 2. ...compare refining anchors using a regression model versus refining them with diffusion, as this would better illustrate the point.
>
> Thank you for this insightful suggestion! We agree that comparing regression‐based refinement against diffusion‐based refinement is a meaningful ablation. To address this, we constructed a regression model by removing the time-embedding component from our denoiser model while keeping the rest of the architecture unchanged, thereby matching the “regression refiner” you proposed.
>
> The results below show that **the anchor regression model performs consistently worse than our BridgeDrive model**. This supports our claim that the iterative, probabilistic refinement provided by the diffusion bridge process in BridgeDrive is essential for achieving higher performance.
>
> | **Metric / Number of Anchors** | **k=20** | **k=40** | **k=60** | **k=80** | **k=200** | **k=500** | **k=1000** |
> |---------------------------|----------|----------|----------|----------|-----------|-----------|------------|
> | **Success Rate (%)**      |          |          |          |          |           |           |            |
> |   BridgeDrive (Ours)      | 72.27    | 73.18    | **74.99**| 72.72    | —         | —         | —          |
> |   Anchor-Only (Ablation)  | 2.72     | 8.18     | 16.81    | 19.54    | 25.92     | 36.81     | 36.36      |
> |   Anchor w/ regression (Ablation)  | 68.18    | 72.72     | 70.91    | 70.91    |  —    | —     | —     |
> | **Driving Score**         |          |          |          |          |           |           |            |
> |   BridgeDrive (Ours)      | 87.02    | 87.24    | **87.99**| 87.27    | —         | —         | —          |
> |   Anchor-Only (Ablation)  | 27.71    | 35.78    | 45.43    | 49.02    | 57.89     | 63.12     | 62.30      |
> |   Anchor w/ regression (Ablation)  |   86.73  | 87.09 | 86.91    | 86.77 |  —    | —     | —     |
>
>
> > 3. On Navsim, the scores for BridgeDrive and DiffusionDrive are similar. Could you try analyzing why?
>
>
> We appreciate the reviewer’s question. We believe the similar performance between BridgeDrive and DiffusionDrive on NAVSIM is largely due to NAVSIM being a **mature, pseudo closed-loop** dataset whose performance is already near **saturation**. **NAVSIM assumes non-reactive agents and no environment–agent interaction** over short horizons, making it comparatively easy and **limiting the room for meaningful improvements**.
>
> For this reason, our paper focuses on the **more realistic closed-loop setting** (Bench2Drive), as reflected in the paper title. Closed-loop evaluation is significantly more challenging because the ego vehicle’s actions influence its own future states as well as those of surrounding agents, creating feedback loops that can amplify small prediction errors over time (Section 2.1). This setting better reflects a policy’s real-world planning ability and has become an emerging standard in the autonomous driving community [2–6].
>
> Our experimental results show that in this challenging and realistic closed-loop setting, BridgeDrive achieves state-of-the-art performance, highlighting the benefits of our diffusion-bridge refinement beyond what can be observed in the saturated NAVSIM benchmark.
>
> [1] Liao, et al. DiffusionDrive: Truncated Diffusion Model for End-to-End Autonomous Driving. CVPR 2025.
>
> [2] Caesar, et al. nuplan: A closed-loop ml-based planning benchmark for autonomous vehicles. arXiv:2106.11810, 2021.
>
> [3] Chitta, et al. Transfuser: Imitation with transformer-based sensor fusion for autonomous driving. TPAMI 2022.
>
> [4] Zheng, et al. Diffusion-based planning for autonomous driving with flexible guidance. ICLR 2025.
>
> [5] Yang, et al. DriveArena: A Closed-loop Generative Simulation Platform for Autonomous Driving. ICCV 2025.
>
> [6] Jia, et al. Bench2Drive: Towards multi-abilitybenchmarkingof closed-loopend-to-endautonomousdriving. NeurIPS 2024.
>
> Thank you again for your valuable comments. We hope that we have adequately addressed all of your concerns.

---

### Official Review · Reviewer_nttY · 2025-10-30

**Soundness:** 2
**Presentation:** 3
**Contribution:** 3
**Rating:** 6
**Confidence:** 5

**Summary:**

The authors present BridgeDrive, which proposes an anchored guided diffusion formulation for closed-loop trajectory planning. BridgeDrive demonstrates SOTA performance on the Bench2Drive benchmark and provide useful insights on parameterization of diffusion models. The only concern is whether the experiments setup is fair for DiffusionDrive( scoring-based v.s predicted anchors), and whether the performance will be highly correlated with how well the anchor prediction is.

**Strengths:**

- Strong empirical SOTA performance on the Bench2Drive benchmark
- Provide good insights on how to parameterize diffusion models output: geomery points verus waypoints

**Weaknesses:**

###

- The main comparison of this work is against DiffusionDrive. The primary comparison is to DiffusionDrive, but the two methods differ significantly: DiffusionDrive scores across *all* anchors and selects the best, while BridgeDrive first classifies one anchor then denoises around it. Please add this additional results for the scoring-based formulation in Table 1 and Table 2
- Why do we care about whether the forward and reverse are symmetric, and how does that affect the performance?  Following the previous points, DiffusionDrive provides a good way for fast inference speed (2 denoising steps compared to 20 steps).
- Couldn’t open the supp videos

**Questions:**

- What if the classified anchors are wrong for BridgeDrive, and how does this affect the perfromance? Does the diffusion models assumes that it always starts from the right anchor?
- Can BridgeDrive, for example, hanlde multiple anchors and then select the best result based on scoring, since this is more practical and may potentailly improve its out-of-distribution capabilities.
- How does the performance degrade by decreasing the diffusion timesteps for Bridgedrive? And why is Diffusiondrive’s sampling speed not ~10x faster than BridgeDrive since diffusion steps are 10x smaller?
- Table 1 miss diffusiondriveo^geo

---

> ### Author Response · Authors · 2025-11-25
> **Response to Reviewer nttY (1/2)**
>
> Thank you for your thoughtful comments! Below, we address your concerns point by point.
>
> > Weakness 1: DiffusionDrive scores across all anchors and selects the best, while BridgeDrive first classifies one anchor then denoises around it. Please clarify classification methods for BridgeDrive and DiffusionDrive.
>
> Thank you for raising this important question. As shown in DiffusionDrive's official implementation [1,2,3], although DiffusionDrive generates trajectories for all anchors, the final trajectory selection is not based on the scores of the generated trajectories. Instead, they employ an **anchor classifier** to select the most likely anchor given the traffic context and use the generated trajectory which corresponds to the selected anchor as the final trajectory. Our trajectory selection procedure effectively follow  the same logic as DiffusionDrive.
>
> [1] https://github.com/hustvl/DiffusionDrive/blob/a3ce7af08a3f2cc5aee656369ed73f50acea3a3a/navsim/agents/diffusiondrive/transfuser_model_v2.py#L490C9-L501C76
>
> [2] https://github.com/hustvl/DiffusionDrive/blob/a3ce7af08a3f2cc5aee656369ed73f50acea3a3a/navsim/agents/diffusiondrive/transfuser_model_v2.py#L545C1-L555C44
>
> [3] https://github.com/hustvl/DiffusionDrive/blob/a3ce7af08a3f2cc5aee656369ed73f50acea3a3a/navsim/agents/diffusiondrive/modules/multimodal_loss.py#L136C9-L157C1
>
> > Weakness 2: Why do we care about whether the forward and reverse are symmetric, and how does that affect the performance? Following the previous points, DiffusionDrive provides a good way for fast inference speed (2 denoising steps compared to 20 steps).
>
>
> The symmetry between the forward and reverse processes is a fundamental requirement for a model to qualify as a diffusion model. Diffusion models generate new samples by reversing the forward noising process; therefore, if the two processes are not symmetric, the model is no longer a true diffusion model. Consequently, it would not benefit from the advantages of diffusion models, nor would it retain the theoretical guarantees that ensure high-quality and diverse sample generation. This point is further supported by our experimental results in Section 4 (Table 2).
>
> Regarding inference speed, we would like to clarify two points. First, the trajectories we generate are much lower-dimensional (20D trajectory + 1D speed) compared to images (>1k dimensions), so the difference between 20 denoising steps and 2 steps has little impact on real wall-clock time (see Table 5, which shows that both methods are suitable for real-time deployment). Second, faster inference is only meaningful when generation quality is comparable. While DiffusionDrive may achieve slightly faster inference in terms of denoising steps, it does not provide the theoretical guarantees or empirical generation quality in the real-world closed-setting that are comparable to our method.
>
> > Question 1:  What if the classified anchors are wrong for BridgeDrive, and how does this affect the performance? Does the diffusion models assumes that it always starts from the right anchor?
>
>
> Thank you for this great question! Even if the classified anchors are incorrect, BridgeDrive can still generate correct trajectories. This is because the diffusion bridge model is capable of producing accurate trajectories even when the anchor is slightly off, as demonstrated in Fig. 1. The figure shows that as long as the classified anchor is not catastrophically wrong, the final trajectory plan remains reasonable. This further emphasize the **robustness** and **safety** advantages of our method.
>
> To further quantify our claim, we performed ablation study by choosing 2nd, 3rd, and 4th most likely anchor according to our anchor classifier, and generate vehicle trajectory from these sub-optimal anchors. The experiment showed that, our BridgeDrive model demonstrates certain resilience against wrong anchor and still achieves a success rate above 60% when 2nd or 3rd most likely anchors are used. However, we can also observe that the success rate drops as less likely anchors are selected.
>
> | anchor-selected | 1st | 2nd | 3rd | 4th |
> | -------- | -------- | -------- |-------- |-------- |
> | **Success Rate (SR) (%)**     | 74.99  | 69.09    | 61.36 | 57.72 |
> | **Driving Score (DS)**     | 87.99  | 85.31    | 80.76 | 77.53 |
>
> > Question 2: Can BridgeDrive, for example, handle multiple anchors and then select the best result based on scoring, since this is more practical and may potentially improve its out-of-distribution capabilities?
>
> Please kindly refer to our responds above to Weakness 1, since our classification follows the same logic as DiffusionDrive.

---

> ### Author Response · Authors · 2025-11-25
> **Response to Reviewer nttY (2/2)**
>
> > Question 3: How does the performance degrade by decreasing the diffusion timesteps for Bridgedrive? And why is Diffusiondrive’s sampling speed not ~10x faster than BridgeDrive since diffusion steps are 10x smaller?
>
> Thank you for raising this question which helps us better clarify our model. In both DiffusionDrive and our BridgeDrive model, the primary computational cost stems from the perception and cross-attention modules. The diffusion process itself is applied only to a small trajectory matrix (e.g., 10x2+1D), contributing a minor portion of the total cost. Consequently, the overall computation time does not increase proportionally with the number of diffusion steps (see Table 5, which shows that both methods are suitable for real-time deployment). We added this explanation in the paper.
>
> > Question 4: Table 1 miss DiffusionDrive^geo.
>
> Thank you for the great suggestion! We initially included DiffusionDrive^geo in Table 2. We have now added it to Table 1 as well.
>
> > Weakness 3: Couldn’t open the supp videos.
>
> We apologize for this incidence. We uploaded new videos of more compatible format.
>
> Thank you again for your valuable comments, which have greatly helped us improve the clarity and quality of our paper. We hope that we have adequately addressed all of your concerns. Please let us know if you have any further questions or concerns.

---

> > ### Comment · Reviewer_nttY · 2025-11-26
> >
> > Thank the authors for the clarification and detailed response, my concerns are all addressed.

---

### Official Review · Reviewer_nA3M · 2025-10-30

**Soundness:** 3
**Presentation:** 3
**Contribution:** 3
**Rating:** 6
**Confidence:** 3

**Summary:**

BridgeDrive introduces a diffusion bridge policy for closed-loop trajectory planning in autonomous driving. Unlike previous diffusion-based planners (e.g., DiffusionDrive) that are mainly tested in open-loop settings, BridgeDrive applies a symmetric diffusion bridge framework connecting expert priors and ground-truth trajectories. This design resolves the theoretical inconsistency of prior truncated diffusion models and enables more stable, consistent, and safety-oriented planning. Experiments on the Bench2Drive benchmark demonstrate state-of-the-art results, surpassing SimLingo by +1.8 in Driving Score and +5% in Success Rate, especially in challenging tasks like merging, overtaking, and traffic sign compliance.

**Strengths:**

1. Recent works utilize diffusion models to enhance autonomous driving planning tasks seems just experiment on open-loop benchmarks (DiffusionPlanner and DiffusionDrive), while this work extend their experiment to close-loop benchmark, which is more chllenging and I think this is a good contribution to the community of autonomous driving.
2. The model BridgeDrive gets SOTA results on close-loop benchmark Bench2Drive.
3. The authors honestly provide the "Limitations and future work" section.

**Weaknesses:**

Theoretical clarity – Some diffusion concepts (e.g., “truncated diffusion,” the exact bridge formulation) are insufficiently explained for general readers.

Fairness of comparison – DiffusionDrive was reimplemented for Bench2Drive, which might affect comparison reliability.

Comfort and smoothness – Prioritizes safety over comfort, leading to frequent braking behavior.

Dependency on LiDAR – Relies on LiDAR input, limiting adaptability to camera-only settings.

**Questions:**

1. Could the author provide detail explain to why "DiffusionDrive that trying to  leverage typical human expert driving behaviors introduces a theoretical inconsistency: its denoising process does not match the forward diffusion process that it is trained on, which diverges from the core principle of diffusion models and can lead to unpredictable behavior and compromised performance." in line 36-43.
2. About Fig1. Intuitively, the startpoint of the denoise process should be noisy, but the waypoints in the leftmost subfigure seem very smooth. So these waypoints are not noise sampled from Gaussian but clustered resulst from expert drivers' behavoir?
3. How do you get the expert prior? Using the data of human drivers' behaviour data on open-loop benchmarks, like nuScenes? Or the "robot expert" trained via reinforcement learning on Bench2Drive?
4. About section 3.2 "DiffusionDrive with Truncated Diffusion". In this section, the author mentioned the asymmetry between the forward "add noise process" and the inverse "denoise process" of DiffusionDrive, whose starting point of forward process is expert prior (anchor) while the endpoint of inverse process is ground truth trajectories. I understand this. But, what does the "truncated" mean? In your paper (line 129-131), it seems that you want to express: timestep t is a pivot, before t, we add some noise, and after t we start denoising, which is really confusing.
5. You method name is "BridgeDrive", where can show "Bridge"? In line 191, you mentioned that you will learn a bridge model $p_\theta(x_t | x_T, z)$, so, you actually bridge the startpoint (ground truth traj) and endpoint (expert prior) at the diffusion process? And the endpoint is not pure gaussion noise traj? And in inference, you directly denoise from expert priors? That sounds plausible.
6. Finally, I think most of the ideas in this paper are based on the weaknesses of DiffusionDrive, so the comparison with DiffusionDrive is important. The problem is, DiffusionDrive only provides its code on NAVSIM and nuScese instead of Bench2Drive. So, to compare with DiffusionDrive, the author can:
  1. Implement your idea based on DiffusionDrive on NAVSIM and nuScenes, then test on the same testset.
  2. Reimplement DiffusionDrive to Bench2Drive, and then test directly, as you have implemented your methods on Bench2Drive.
  The author chooses the second method and although they provide detailed implementation details of how to reimplement DiffusionDrive to Bench2Drive, the fact is reimplement a method from one benchmark to another is challengeing, we cannot guarantee that during reimplementation, there maybe some errors lead to performance drop. So, I recommend the author can adopt the first compare method, in that way, DiffusionDrive have offical code and you can easily implement your method on it and avoid cross benchmark implement. And the result will be more convincing.

---

> ### Author Response · Authors · 2025-11-25
> **Response to Reviewer nA3M (1/2)**
>
> Thank you for your encouraging comments! Below, we address your concerns point by point.
>
> > Question 1 & Weakness 1: Could the authors provide detail explain to why "DiffusionDrive that trying to leverage typical human expert driving behaviors introduces a theoretical inconsistency..." in line 36-43?
>
> A fundamental requirement for a model to qualify as a diffusion model is that the forward process ($x_0 \to x_T$) and the reverse process ($x_T \to x_0$) must be symmetric. This symmetry is essential because diffusion models generate new samples by reversing the forward noising process. In DiffusionDrive, however, the forward process defines $x_0$ as **anchor** and $x_T$ as noisy anchor, while the reverse process defines $x_0$ as **final trajectory plan** and $x_T$ as noisy anchor. This mismatch of $x_0$ in the two processes means that they are not symmetric, leading to a theoretical inconsistency.
>
> > Question 2 & Weakness 1: About Fig 1. Intuitively, the startpoint of the denoise process should be noisy, but the waypoints in the leftmost subfigure seem very smooth. So these waypoints are not noise sampled from Gaussian but clustered resulst from expert drivers' behavoir?
>
> Yes, the starting point in our model is anchor rather than noise, and this actually highlights the main difference between **diffusion bridge models** [1] and standard diffusion models. In standard diffusion models, the diffusion process maps between a data distribution and a Gaussian noise distribution. In diffusion bridge models, it maps between two arbitrary distributions, making it more general, since setting one of the distributions to Gaussian recovers the standard diffusion process. In the context of BridgeDrive, our model maps between anchors and trajectory plans, effectively acting as a tool to shape coarse anchor trajectories into refined, high-quality plan trajectories.
>
> [1] Zhou, et al. Denoising diffusion bridge models. ICLR 2024.
>
> > Question 3: How do you get the expert prior?
>
> As detailed in Section 3.1, we cluster expert driving trajectories from the Bench2Drive training set and use the resulting cluster centers as typical expert priors.
>
> > Question 4 & Weakness 1: About section 3.2 "DiffusionDrive with Truncated Diffusion"... What does the "truncated" mean?
>
> In a standard diffusion model, the forward process gradually adds noise to the data until it is **completely corrupted** into pure Gaussian noise. In contrast, DiffusionDrive only adds **a small amount of noise** during its forward process, which can be viewed as a **truncation** of the standard forward diffusion process. We took the term "truncated" from the original DiffusionDrive paper.
>
> > Question 5: You method name is "BridgeDrive", where can show "Bridge"?
>
> The term "Bridge" refers to our use of a diffusion bridge model [1] to __bridge__ expert anchors and the final trajectory plans. This provides a principled diffusion framework for trajectory planning and resolves the theoretical inconsistency present in DiffusionDrive.
>
> [1] Zhou, et al. Denoising diffusion bridge models. ICLR 2024.

---

> ### Author Response · Authors · 2025-11-25
> **Response to Reviewer nA3M (2/2)**
>
> > Question 6 & Weakness 2: Implement BridgeDrive on NAVSIM, then test on the same testset.
>
> Thank you for the suggestion! We conducted additional experiments for our method on the **NAVSIM** dataset. Following the setup of DiffusionDrive [1], we evaluate our model using the PDM score (PDMS), a comprehensive metric integrating no at-fault collisions (NC), drivable area compliance (DAC), time-to-collision (TTC), comfort (Comf.), and ego progress (EP).
>
>
> | Method | NC | DAC | TTC | Comf. | EP | PDMS |
> | -------- | -------- | -------- |-------- |-------- |-------- |-------- |
> | **DiffusionDrive (reported in [1])** | 98.2 | 96.2 | 94.7 | 100 | 82.2| 88.1 |
> | **DiffusionDrive (reproduced with different random seed)** | 98.2  | 95.9 | 94.3 | 100 | 81.9 | 87.6 |
> | **BridgeDrive (ours)** | 98.2 | 96.1 | 94.5 | 100 | 82.3 | 88.0 |
>
> The results in the above table demonstrate that our BridgeDrive method attains competitive performance on the NAVSIM benchmark. We posit that this performance is notable, as NAVSIM is a mature dataset, where existing state-of-the-art models have already **saturated** its performance potential (NAVSIM is after all a psuedo closed-loop dataset with non-reactive agents under a strong assumption of NO environment and agent interaction over a short horizon).
>
> For this reason, our paper primarily targets the **more realistic closed-loop setting** (i.e., Bench2Drive), as emphasized in the paper title. The closed-loop setting is more challenging and better reflects a policy’s real-world planning capability, since the ego vehicle’s decisions influence both its own future states and those of surrounding agents. This interaction creates a feedback loop that can amplify small prediction errors over time, as discussed in Section 2.1. We also highlight that evaluating in closed-loop settings has become an emerging trend in the autonomous driving community [2,3,4,5], driving more realistic assessments that better reflect actual technological progress. Our experimental results demonstrated that our BridgeDrive method outperformed previous state-of-the-art methods in this challenging setting.
>
> [1] Liao, et al. DiffusionDrive: Truncated Diffusion Model for End-to-End Autonomous Driving. CVPR 2025.
>
> [2] Caesar, et al. nuplan: A closed-loop ml-based planning benchmark for autonomous vehicles. arXiv:2106.11810, 2021.
>
> [3] Chitta, et al. Transfuser: Imitation with transformer-based sensor fusion for autonomous driving. TPAMI 2022.
>
> [4] Zheng, et al. Diffusion-based planning for autonomous driving with flexible guidance. ICLR 2025.
>
> [5] Yang, et al. DriveArena: A Closed-loop Generative Simulation Platform for Autonomous Driving. ICCV 2025.
>
>
> > Weakness 3: Comfort and smoothness – Prioritizes safety over comfort, leading to frequent braking behavior.
>
> In the paper, we acknowledged that we prioritized safety over comfort in the caption of Table 2. This is because we believe that safety must come first - it only makes sense to pursue high comfort levels once safety is assured.
>
> > Weakness 4: Dependency on LiDAR – Relies on LiDAR input, limiting adaptability to camera-only settings.
>
> In the paper, we acknowledged this limitation in Section 5. However, we would like to emphasize that our BridgeDrive framework is **model-agnostic**, meaning it is **fully compatible** with camera-only model architectures as well. In this paper, we opted to use LiDAR input because we adopted the model architecture from DiffusionDrive [1] for a fair comparison, and that architecture requires LiDAR input. As noted in Section 5, we leave the exploration of camera-only models for future work.
>
> [1] Liao, et al. DiffusionDrive: Truncated Diffusion Model for End-to-End Autonomous Driving. CVPR 2025.
>
> Thank you again for your thoughtful comments, which have greatly helped us improve the clarity and quality of our paper. We hope that we have adequately addressed all of your concerns. Please let us know if you have any further questions or concerns.

---

> ### Comment · Reviewer_nA3M · 2025-11-26
>
> Thanks to the authors for conducting the additional experiments on Navsim. These results further strengthen the evidence of BridgeDrive's improvements over DiffusionDrive and help address my earlier concerns. I am satisfied with the revisions and will maintain my positive evaluation.

---

### Author Response · Authors · 2025-11-25
**General response**

We thank all reviewers for their valuable feedback and constructive comments, which have helped us significantly improve the clarity and quality of our manuscript. We are encouraged that **all reviewers recognized our work as a novel and important contribution to the field**. The key points of their feedback are summarized below:

* **Reviewer nA3M** and **Reviewer nttY** affirmed that our SOTA performance in closed-loop evaluation with reactive agents constitutes a valuable contribution to the autonomous driving community.

* **Reviewer RMqE** appreciated our identification of the theoretical inconsistency in DiffusionDrive, an issue that has misled the community, and endorsed our BridgeDrive model as a reasonable solution.

* **Reviewer Awhw** acknowledged that our proposed BridgeDrive model establishes a complete theoretical framework for connecting anchor data to trajectory generation, thereby addressing the heuristic shortcomings of previous methods, while maintaining practicality through real-time inference and robust multi-modal trajectory generation, as well as achieving notable improvements on Bench2Drive and demonstrating strong closed-loop performance benefits.

We have addressed each reviewer's concerns separately below their respective review. We have updated our manuscript to include some additional results and details requested by the reviewers, with changes highlighted in orange:

1. We performed an ablation study to **isolate the individual contributions of the diffusion bridge model and the anchor prior (Table 9)**, confirming that BridgeDrive's SOTA performance stems from the synergistic effect of both components.

2. We conducted **ablation study on anchor diversity and classifier accuracy (Tables 8 and 9)**, concluding that achieving an optimal performance requires a careful balance between these two factors.

3. We additionally **compared our method with anchor-based regression method (Table 9)** to support our claim that the iterative, probabilistic refinement provided by the diffusion bridge process in BridgeDrive is essential for achieving higher performance.

4. We extended our experiments to the **NavSim dataset (Table 6)**, demonstrating that our proposed solution generalizes effectively to other benchmarks and outperforms existing anchor-based SOTA methods (VADv2 and Hydra-MDP).

---

### Author Response · Authors · 2025-12-03
**Summary of Rebuttal**

Dear Area Chair,

Thank you for your effort to handle our submission! We are writing to provide a summary of each reviewer's main comments and our responses.

### **Reviewer nA3M**

- **Positive feedback**: "The model BridgeDrive gets SOTA results on close-loop benchmark Bench2Drive."
- **Main request**: Evaluate BridgeDrive on another benchmark NAVSIM.
- **Our response**: We conducted new experiments on NAVSIM (Table 6), demonstrating BridgeDrive's competitve performance on this benchmark. We further explained that the performance potential on NAVSIM has been saturated as it's a psuedo closed-loop dataset with non-reactive agents under a strong assumption of no environment and agent interaction over a short horizon. Hence, the more realistic closed-loop setting (Bench2Drive) considered in our paper is a better benchmark for evaluating real-world driving ability.
- **Reviewer's acknowledgement**: "These results further strengthen the evidence of BridgeDrive's improvements over DiffusionDrive and help address my earlier concerns. I am satisfied with the revisions and will maintain my positive evaluation."

### **Reviewer nttY**

- **Positive feedback**: "Strong empirical SOTA performance on the Bench2Drive benchmark"
- **Main requests**: Clarify the setup and robustness of the anchor classifier, and the diffusion inference speed of BridgeDrive.
- **Our response**: We clarified that our anchor classification setup was identical to DiffusionDrive to ensure a fair comparison, and conducted additional ablation study to show that our model remains resilient even when forced to use suboptimal anchors (Table 8). We explained that our model is suitable for real-time deployment (Table 5), as the diffusion step count has negligible impact on wall-clock time due to the low-dimensionality nature of trajectory data.
- **Reviewer's acknowledgement**: "my concerns are all addressed."

### **Reviewer RMqE**

- **Positive feedback**: BridgeDrive identifies the theoretical inconsistency in DiffusionDrive which has misled the community, and proposes a reasonable solution to solve this issue.
- **Main requests**: Clarify the use of geometric waypoints vs temporal waypoints, and the choice of Think2Drive expert vs PDM-Lite expert. Demonstrate the necessity of diffusion-based planner by comparing it with pure anchor-based methods.
- **Our response**: We clarified that our claim about the superiority of geometric waypoints is within the Bench2Drive benchmark, which is supported by our experiment results in Table 2 and prior methods in the literature (e.g., Transfuser++). We explained that the selection of expert is a result of specific design of each baseline method (e.g., Think2Drive has many surrounding views, while PDM-Lite only has a single front view). We performed new ablation study to compare BridgeDrive with the anchor classification method and anchor-based regression method (Table 6 and Table 9). Our results showed that BridgeDrive consistently outperformed these anchor-based methods, demonstrating the necessity of diffusion-based planners.
- **Reviewer's acknowledgement**: We had addressed all of Reviewer RMqE's initial concerns. The reviewer asked three follow-up questions, but they had not acknowledged our response to those follow-up questions before the discussion was frozen.

### **Reviewer Awhw**

- **Positive feedback**: BridgeDrive establishes a complete theoretical framework for connecting anchor data to trajectory generation, thereby addressing the heuristic shortcomings of previous methods, while maintaining practicality through real-time inference and robust multi-modal trajectory generation, as well as achieving notable improvements on Bench2Drive and demonstrating strong closed-loop performance benefits.
- **Main requests**: Evaluate BridgeDrive on another benchmark NAVSIM. Perform sensitivity analysis for anchors. Analyze the influence of scene context versus anchor selection on the final trajectory.
- **Our response**: We performed the requested additional evaluation on NAVSIM (Table 6, see our summary for Reviewer nA3M above). We conducted sensitivity analysis regarding the number of anchors and anchor classification accuracy (Table 8 and Table 9). We conducted extra ablation study to isolate the individual contributions of anchors and scenes (Table 6 and Table 9), demonsrating that the anchor and scene jointly shape the generated trajectory.
- **Reviewer's acknowledgement**: "BridgeDrive truly provides a complete and empirical theoretical framework for combining anchor-based methods with diffusion. Now I am satisfied with the paper and have raised my score."

---

### Meta-Review · Area_Chair_Fyr1 · 2025-12-06

**Summary:**

This paper introduces BridgeDrive, which adapts the Denoising Diffusion Bridge Model to generate geometric path waypoints from an anchor distribution. By enforcing a symmetric and theoretically consistent forward–reverse process, BridgeDrive improves closed-loop performance.

The overall opinions from reviewers are leaning towards acceptance. The authors have provided a series of new results to address the reviewers' concerns. I'm inclined to accept the paper; however, I also encourage the authors to provide results on other benchmarks as well as sufficient ablations in their final version.

**Reviewer Concerns:**

Some concerns that I think are still not well-addressed:
- Presentation issues (Reviewer nA3M)
- Most of the ideas are based on the weaknesses of DiffusionDrive, but not tested on the same benchmeark (Reviewer nA3M, RMqE)
- Concern of only testing on the Bench2Drive benchmark (Reviewer nA3M, RMqE, Awhw, AC). The authors have provided new results on NAVSIM during rebuttal, but the proposed method does not seem to provide meaningful performance improvement over DiffusionDrive.
- Needs additional ablation (Reviewer RMqE, Awhw)


Some comments that I believe have been addressed or partly addressed during rebuttal, based on additional experimental results:
- Doubts regarding using bridge diffusion (Reviewer nA3M, nttY)
- Needs additional comparison with DiffusionDrive and other SOTA anchor-based methods (Reviewer nttY, RMqE)

**Reviewer Scores:**

Reviewer nttY and Awhw mentioned that their concerns are addressed, and might consider increasing the score.

Reviewer nA3M mentioned that it will not change the score.

---

### Decision · Program_Chairs · 2026-01-26

Accept (Poster)